# ONE MODEL TO CRITIQUE THEM ALL: REWARDING AGENTIC TOOL-USE VIA EFFICIENT REASONING

## ABSTRACT

Reward models (RMs) play a critical role in aligning large language models (LLMs) with human preferences. Yet in the domain of tool learning, the lack of RMs specifically designed for function-calling tasks has limited progress toward more capable agentic AI. We introduce TOOLRM, a family of lightweight generative RMs tailored for general tool-use scenarios. To build these models, we propose a novel pipeline that constructs pairwise preference data using rule-based scoring and multidimensional sampling. This yields *ToolPref-Pairwise-30K*, a diverse, balanced, and challenging dataset of critique tasks that supports reinforcement learning with verifiable feedback. To evaluate tool-use RMs, we also introduce TRBENCH$_{\text{BFCL}}$, a benchmark built on the agentic evaluation suite BFCL. Trained on our constructed data, models from the Qwen3-4B/8B series achieve up to 14.28% higher accuracy, substantially outperforming frontier models such as Claude 4 and OpenAI o3 in pairwise reward judgments. Beyond training objectives, TOOLRM generalizes to broader critique tasks, including Best-of-N sampling and self-correction. Experiments on ACEBENCH highlight its effectiveness and efficiency, enabling inference-time scaling and reducing output token usage by over 66%. We release data and model checkpoints to facilitate future research.

## 1 INTRODUCTION

Recent advances in agentic artificial intelligence (AI) have been driven in large part by the tool-use capabilities of large language models (LLMs) (Schick et al., 2023; Patil et al., 2024; OpenAI, 2025). By leveraging external tools, LLMs can recognize their limitations and extend their capabilities through environment interaction. The research focus has recently shifted from behavior cloning via supervised finetuning on curated trajectories (Schick et al., 2023; Tang et al., 2023) to trial-and-error approaches based on reinforcement learning from verifiable rewards (RLVR) (Feng et al., 2025; Qian et al., 2025), enabling more generalizable and robust tool-use behavior.

Despite these gains, the lack of reliable reward models (RMs) tailored to tool-use tasks remains a core limitation. Most existing methods depend on verified tool-call trajectories for feedback, which restricts scalability to domains lacking such annotations. At inference time, the absence of precise reward signals also makes it hard to leverage multiple sampled answers for test-time selection (Wang et al., 2023; Snell et al., 2025). We argue that developing a robust RM—capable of evaluating tool-use behavior without requiring ground-truth labels—is critical for advancing this field.

Designing effective RMs for tool-use presents three key challenges: **(C1)** Constructing high-quality preference pairs that reflect tool-use intent (Liu et al., 2024a). **(C2)** Enabling generalizable critique beyond 3H-style modeling (Askell et al., 2021), as tool-use tasks often allow more objective, causal reasoning. **(C3)** Evaluating RM performance in this setting, which remains underexplored for both frontier LLMs and specialized critics.

To address these challenges, we introduce TOOLRM, a family of lightweight generative RMs for general tool-use tasks. We design a two-stage pipeline to construct high-quality preference data. First, we curate and validate tool-calling trajectories from diverse open-source datasets, segment them into context–response pairs, and sample alternative responses using multiple LLMs. Instead of relying on ground-truth matches, we apply rule-based labeling to capture fine-grained preferences. A multidimensional sampling strategy ensures diverse scenarios, varied preference intensity, and high task complexity **(C1)**. To strengthen critique ability, we train TOOLRM with a pairwise objective

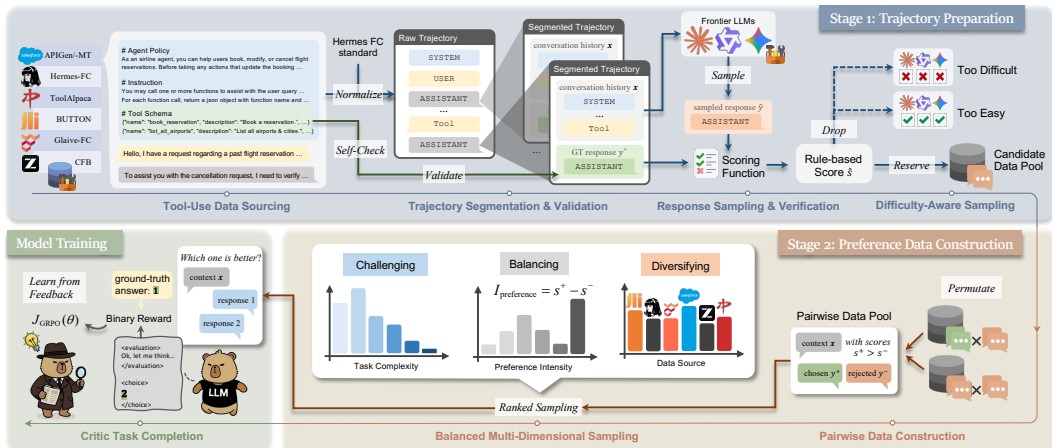

Figure 1: Overview of the proposed pipeline for training ToolRM.

using unified instructions and verifiable supervision, enabling the model to learn robust reasoning without curated traces **(C2)**. We also introduce $\text{TRBENCH}_{\text{BFCL}}$, a benchmark based on BFCL (Patil et al., 2025), to systematically evaluate RM performance on tool-use tasks **(C3)**.

In summary, our key contributions are as follows:

- We propose a novel pipeline for generating high-quality pairwise preference data for reward modeling in tool-use scenarios. Using seven open-source tool-calling datasets, we construct *ToolPref-Pairwise-30K*, a diverse and balanced set of 30,000 challenging preference pairs. This resource is publicly released to support future work in tool-oriented reward modeling.

- We train $\text{TOOLRM}$ on the Qwen3-4B/8B series using RLVR, achieving strong gains in pairwise reward judgments. Beyond training objectives, our models generalize to broader critique tasks, enabling efficient inference-time scaling and producing compact, informative critiques.

- We introduce $\text{TRBENCH}_{\text{BFCL}}$, a dedicated benchmark for evaluating reward models in tool-use settings. Our analysis reveals that even state-of-the-art LLMs and specialized reward models show significant gaps on this benchmark, underscoring the need for targeted solutions.

## 2 METHODOLOGY

We introduce a pipeline for training a generative reward model for tool-use. As shown in Figure 1, we first label tool-calling trajectories using rule-based verifiers. In stage two, we construct pairwise preferences via balanced multidimensional sampling. The model is trained with a pairwise critique objective under the RLVR paradigm, resulting in ToolRM with strong evaluative capabilities.

### 2.1 TRAJECTORY PREPARATION

**Task Sourcing.** To build a diverse dataset, we collate function-calling tasks from seven open-source, tool-learning datasets, spanning a wide variety of task domains and trajectory patterns: APIGen (Liu et al., 2024b), APIGen-MT (Prabhakar et al., 2025), BUTTON (Chen et al., 2025b), ComplexFuncBench (Zhong et al., 2025), Hermes-Function-Calling (Teknium et al., 2025), Glaive-Function-Calling[1], and ToolAlpaca (Tang et al., 2023). To address format inconsistencies across these sources, we standardize all conversation records of raw tasks into format-aligned trajectories $\mathcal{T}_{\text{raw}} = \{\tau_i\}_{i=1}^{N}$, discarding any data with invalid role orders. The message format within each trajectory $\tau_i$ is normalized to adhere to the Hermes Function Calling standard[2], where special tags `<tools>`, `<tool_call>`, and `<tool_response>` are used to enclose tool schemas, calls, and responses, respectively. At the beginning of each $\tau_i$, a function-calling prompt is uniformly included

---

[1]We use a 5k cleaned glaive-function-calling subset in hermes-function-calling-v1.
[2]https://github.com/NousResearch/Hermes-Function-Calling

Table 1: Statistics for each constituent dataset. *Raw* and *Filtered* are reported by the number of original tasks, while *Segmented* counts the number of segmented trajectories, with *Avg. T* indicating their average number of turns. Trajectory patterns in each dataset are characterized from turn, step, and order perspectives: 'ST' and 'MT' denote 'single-turn' and 'multi-turn'; 'SS' and 'MS' denote 'single-step' and 'multi-step'; 'P' and 'S' denote 'parallel' and 'sequential', respectively.

| Data Source | Raw | Filtered | Segmented | Avg. T | Schemas | Pattern of Trajectory | | | Task Domain |
|---|---|---|---|---|---|---|---|---|---|
| | | | | | | Turn | Step | Order | |
| APIGen | 60,000 | 60,000 | 59,960 | 3.00 | 4,205 | ST | SS/MS | P | Finance/Sports/Technology/Travel ... |
| APIGen-MT | 5,000 | 4,874 | 20,055 | 11.75 | 26 | MT | SS/MS | P/S | Airline/Retail |
| BUTTON | 8,000 | 8,000 | 20,811 | 5.19 | 22,101 | MT | SS/MS | P/S | Daily Life |
| ComplexFuncBench | 1,000 | 1,000 | 3,259 | 5.43 | 40 | ST | MS | S | Hotel/Flight/Attraction/Car Rental/Taxi |
| Glaive-Function-Calling | 5,209 | 4,344 | 6,747 | 4.82 | 1,565 | MT | SS/MS | P | Stocks and Orders/Movie/Flight Services ... |
| Hermes-Function-Calling | 1,893 | 1,724 | 1,724 | 3.00 | 2,383 | ST | SS/MS | P | Information Extraction/API Call/Software ... |
| ToolAlpaca | 4,098 | 2,510 | 6,194 | 4.24 | 2,040 | ST | SS/MS | P/S | News/Jobs/Finance/Entertainment ... |

as the system message, along with the schemas of available tools in the task. Additional agent policies are prepended to this message for complex tasks from specific sources (e.g., APIGen-MT). See Appendix F for an example of a tool-use task trajectory.

**Trajectory Segmentation and Validation.** To enable subsequent rule-based verification of arbitrary trajectories against ground-truth answers, we first perform tool schema validation for each trajectory $\tau_i$. Tool schemas are typically provided as dictionary objects, which we verify as valid JSON schemas describing tools compatible with OpenAI's tool-calling format[3]. Invalid schemas are corrected, and duplicates are removed. The validated schemas are then wrapped into function-type JSON objects and incorporated into the aforementioned system message as tool descriptions.

Next, we partition each raw trajectory $\tau_i \in \mathcal{T}_{\text{raw}}$ into sub-trajectories that each terminate with an assistant message. This yields a set of segmented trajectories, denoted as $\mathcal{T}_{\text{seg}} = \{\tau_j\}_{j=1}^{M}$. Each segment $\tau_j$ consists of a conversation history $\mathbf{x}_j$ (the sequence of messages preceding the assistant message) and its corresponding assistant response $y_j$. A preliminary filtering is then applied: we retain a segment $\tau_j$ only if the message following $y_j$ in the raw trajectory $\tau_i$ does not contain any unsuccessful tool response, which ensures the basic validity of tool calls in $y_j$.

A stricter validation of tool calls is further employed for the assistant response within each retained trajectory $\tau_j$. Each tool call in $y_j$ is validated against the tool schemas: it must be parsable in the required format (e.g., `{"name":"...","arguments":{...}}`) and its function name and arguments must match the schema definitions. Responses containing duplicate tool calls are also discarded. Finally, only the trajectories $\tau_j = (\mathbf{x}_j, y_j)$ that pass all format and content checks are kept. For these validated trajectories, we designate the response within them as the ground-truth response $y_j^*$, and the clean dataset consists of these validated pairs $\mathcal{T}_{\text{clean}} = \{(\mathbf{x}_j, y_j^*)\}_{j=1}^{M'}$. Table 1 summarizes statistics for each data source, including the number of unique tool schemas and the distribution of tool-call trajectory patterns, measured by turn-, step-, and order-wise occurrences.

**Response Sampling and Verification.** In this phase, we begin by sampling multiple model responses for each conversation history. To ensure diversity in the outputs, we select five models from three different families with varying tool-calling capabilities: Claude-3.7-Sonnet, Gemini-2.5-Pro, Qwen2.5-Max, Qwen3-32B, and Qwen3-8B. For each pair $(\mathbf{x}_j, y_j^*)$ in the cleaned dataset $\mathcal{T}_{\text{clean}}$, the context $\mathbf{x}_j$ is sent to all five models, yielding a set of new assistant responses $\{\hat{y}_{j,k}\}_{k=1}^{5}$. Each sampled response $\hat{y}_{j,k}$ is then scored using a rule-based function that compares it against its corresponding ground-truth response $y_j^*$, yielding a score between 0 and 1. Unlike prior rule-based TIR approaches (Qian et al., 2025), our method for training the reward model prioritizes the correctness of tool call content (reasoning ability) over strict format adherence (instruction-following ability), since downstream applications often use varying tool call structures. Consequently, we only score $\hat{y}$ that can be successfully parsed into the expected tool-call format and discard all others.

For a given ground-truth response $y^*$ and a sampled response $\hat{y}$ (we drop indices $j, k$ for simplicity), let $\mathcal{C}^* = \{c_i^*\}_{i=1}^{N_G}$ and $\hat{\mathcal{C}} = \{\hat{c}_l\}_{l=1}^{N_P}$ denote the lists of tool calls parsed from them, respectively. Each tool call is a JSON object containing a string-typed `name` and a dictionary of `arguments`. Scoring starts with two disqualifiers: if either applies, the final score $\hat{s}$ is set to 0:

---

[3] https://platform.openai.com/docs/guides/function-calling

- *Mismatched Number of Tool Calls.* The number of predicted tool calls does not match the number of ground-truth tool calls:

$$|\hat{\mathcal{C}}| \neq |\mathcal{C}^*| \Rightarrow \hat{s} = 0 \tag{1}$$

- *Duplicated Tool Calls.* The set of predicted tool calls contains identical duplicates (both name and arguments are the same). For $\hat{c}_l, \hat{c}_m \in \hat{\mathcal{C}}$:

$$\exists l \neq m \text{ s.t. } \texttt{is\_identical}(\hat{c}_l, \hat{c}_m) \Rightarrow \hat{s} = 0 \tag{2}$$

If a sampled response $\hat{y}$ passes the above initial checks, a match score $s_i$ is calculated for each ground-truth tool call $c_i^* \in \mathcal{C}^*$. This score is determined by matching $c_i^*$ with the predicted tool call of the same name that achieves the highest argument similarity. Specifically:

$$s_i = \max_{\hat{c} \in \hat{\mathcal{C}}} \mathbb{1}[c_i^*.\texttt{name} = \hat{c}.\texttt{name}] \cdot \texttt{sim}(c_i^*.\texttt{arguments}, \hat{c}.\texttt{arguments}) \tag{3}$$

where $\mathbb{1}[\cdot]$ is an indicator function equal to 1 if the tool names match and 0 otherwise. This ensures that arguments are only compared when tool names align. The argument similarity function $\texttt{sim}(\cdot)$ measures the ratio of identical key-value pairs to the total number of unique keys across both dictionaries. A key-value pair is considered identical only if the key appears in both dictionaries and the corresponding values match, with string comparisons performed in a case-insensitive manner. If both dictionaries are empty, the similarity is defined as 1. The final rule-based score $\hat{s}$ can then be calculated as the mean of all individual match scores $s_i$, with $\hat{s} = 1$ when both $y^*$ and $\hat{y}$ contain no tool calls:

$$\hat{s} = \frac{1}{N_{\text{G}}} \sum_{i=1}^{N_{\text{G}}} s_i \tag{4}$$

**Difficulty-Aware Down-Sampling.** After collecting all rule-based scores for sampled responses, we perform difficulty-aware down-sampling. This is done by grouping all sampled responses by their original context $\mathbf{x}_j$. Empirically, tasks that are either too easy or too difficult are not ideal for model training: (1) contexts for which all sampled responses have a rule-based score of 1 are discarded, as they offer no meaningful variation for model critique; (2) contexts for which no sampled response receives a rule-based score of 1 are also removed, as such cases likely contain noise in either $\mathbf{x}_j$ or $y_j^*$. We retain the remaining candidate data as a flat set of quadruples:

$$\mathcal{D}_{\text{cand}} = \{(\mathbf{x}_j, y_j^*, \hat{y}_{j,k}, \hat{s}_{j,k}) \mid \text{context } j \text{ passes the filter}\} \tag{5}$$

Each contains the conversation history, the ground-truth response, a sampled response, and the corresponding rule-based score. This pool serves as the source for constructing preference datasets.

## 2.2 PREFERENCE DATA CONSTRUCTION

**Pairwise Data Construction.** This section outlines the construction of data for training RM as a critic. Such models are typically used to evaluate data in either a *pointwise* or *pairwise* manner. Our preliminary experiments with a pointwise model, using rule-based scores as supervision signals, led to superficial overfitting. The model learned to mimic the score distribution in the training set rather than develop genuine analytical skills—a form of reward hacking that limited its performance on out-of-distribution (OOD) tasks. To address this limitation, we focus on training reward models with pairwise critique tasks, which mitigate the above issue by relying on comparative judgments rather than direct scoring. The pairwise reward model is designed to distinguish a preferred response from a rejected one for a given context. To construct the training data for this, we sample pairs of responses from the preprocessed data pool $\mathcal{D}_{\text{cand}}$, where ground-truth preferences are determined by their rule-based scores. Each pair consists of a chosen response $y^+$ and a rejected response $y^-$ that shares the same context but differs in score. We traverse $\mathcal{D}_{\text{cand}}$ and arrange the permutations according to the above rules to get a candidate pairwise data pool:

$$\mathcal{D}_{\text{pair-cand}} = \{(\mathbf{x}, y^*, y^+, y^-, s^+, s^-) \mid s^+ > s^-, (\mathbf{x}, y^*, y^+, s^+), (\mathbf{x}, y^*, y^-, s^-) \in \mathcal{D}_{\text{cand}}\} \tag{6}$$

**Balanced Multi-Dimensional Sampling.** To enable efficient training with fewer data, we then adopt a balanced, multi-dimensional sampling strategy to select samples from $\mathcal{D}_{\text{pair-cand}}$. In this strategy, we focus on the following three dimensions of data:

- *Diversity of Data Sources.* Incorporating a diverse range of tool schemas and user queries enhances the generalizability of trained models. To this end, we aim to sample contexts from different sources in a balanced manner. For each context $\mathbf{x}$ in data, we denote its source as $\mathbf{x}$.`source`.

- *Coverage of Preference Intensity.* For each pair of chosen and rejected responses, the difference in their rule-based scores reflects the intensity of the preference signal: a large difference signifies a strong preference, while a small difference suggests a weak one. To train a more robust reward model, our data sampling process is designed to cover this full spectrum of preference signals, from weak to strong. For each pairwise data point, we measure its preference intensity by:

$$I_{\text{preference}} = s^+ - s^- \tag{7}$$

- *Complexity of Tasks.* Challenging the reward model with more complex tasks is essential for enhancing its analytical capabilities. We calculate the complexity score of one candidate data point according to its ground-truth response $y^*$:

$$S_{\text{complex}} = |\mathcal{C}^*| + \sum_{i=1}^{N_{\text{G}}} |c_i^*.\texttt{arguments}| \tag{8}$$

where $\mathcal{C}^*$ is the set of tool calls parsed from $y^*$. Both the number of tool calls and arguments are accumulated to measure the task complexity. Notably, over-complicated data points ($S_{\text{complex}} > 50$) are filtered out for a higher success rate of rollout trajectory in the model training stage.

Guided by the above principles, we use a heuristic algorithm to select samples from $\mathcal{D}_{\text{pair-cand}}$ that are more efficient for model training. Specifically, we prioritize samples with higher complexity scores $S_{\text{complex}}$ while ensuring that the data source $\mathbf{x}$.`source` and preference intensity $I_{\text{preference}}$ are as balanced as possible, resulting in a subset of pairwise data $\mathcal{D}_{\text{pair-sampled}} \subseteq \mathcal{D}_{\text{pair-cand}}$ for subsequent model training. Details of the heuristic algorithm are provided in Appendix E.

## 2.3 MODEL TRAINING

**Critique Task Design.** To elicit the evaluative capabilities of models as critics, we prompt them as expert AI performance evaluators. Given a conversation history and two candidate assistant responses, their task is to provide a thorough evaluation of each and then select the superior one, outputting its name within `<choice>` tags. We tailor instructions to different models according to their native output style: reasoning models follow a *think-mode* template, where their evaluations are embedded within the reasoning process, while non-reasoning models use a *no-think-mode* template, explicitly presenting their evaluations within `<evaluation>` tags. To ensure consistent and comprehensive critiques, we further establish unified evaluation criteria that guide the model. These guidelines specify which types of errors in tool-invocation responses should be penalized. For each sampled data $(\mathbf{x}, y^*, y^+, y^-, s^+, s^-) \in \mathcal{D}_{\text{pair-sampled}}$, we format the conversation history $\mathbf{x}$ into a single string. This string is then concatenated with the two assistant responses $y^+$ and $y^-$ to form the final input query $q$. To reduce position bias and prevent reward hacking during training, we randomly swap the order of the assistant responses in 50% of the queries, recording the position of $y^+$ as the ground-truth answer $a$. The resulting dataset $\mathcal{D}_{\text{pref}} = \{(q, a)_i\}_{i=1}^K$ is then used to train the reward model. Please see Appendix G for detailed prompt templates.

**Training Objective.** We train the target reward model within the RLVR paradigm using Group Relative Policy Optimization (GRPO) (Shao et al., 2024), a variant of Proximal Policy Optimization (PPO) (Schulman et al., 2017) that improves efficiency and reduces computational cost by replacing the critic network with grouped relative advantages. Given an input query $q$ and its ground-truth answer $a$, let $\mathcal{O} = \{o_1, o_2, \ldots, o_G\}$ denote the set of rollout trajectories generated by the old policy $\pi_{\theta_{\text{old}}}$. Our goal is to optimize the policy $\pi_\theta$ by maximizing the following objective:

$$J_{\text{GRPO}}(\theta) = \mathbb{E}_{(q,a)\sim\mathcal{D}_{\text{pref}}, \{o_i\}_{i=1}^G \sim \pi_{\theta_{\text{old}}}(\cdot|q)}$$

$$\left[ \frac{1}{G} \sum_{i=1}^{G} \frac{1}{|o_i|} \sum_{t=1}^{|o_i|} \left[ \min\left( \frac{\pi_\theta(o_{i,t}|q, o_{i,<t})}{\pi_{\theta_{\text{old}}}(o_{i,t}|q, o_{i,<t})} A_{i,t}, \text{clip}\left( \frac{\pi_\theta(o_{i,t}|q, o_{i,<t})}{\pi_{\theta_{\text{old}}}(o_{i,t}|q, o_{i,<t})}, 1-\epsilon, 1+\epsilon \right) A_{i,t} \right) \right] \right] \tag{9}$$

where $\epsilon$ is a clipping-related hyper-parameter for stabilizing training. $A_{i,t}$ denotes the relative advantage calculated based on outputs of each rollout group:

$$A_{i,t} = \frac{r_i - \text{mean}(\{r_1, r_2, \ldots, r_G\})}{\text{std}(\{r_1, r_2, \ldots, r_G\})} \tag{10}$$

Here, $r_i$ denotes the binary reward assigned to the rollout trajectory $o_i$. It is determined by whether a valid choice can be successfully extracted from $o_i$ and whether it accurately answers $q$:

$$r_i = \begin{cases} 1, \text{if is\_equivalent}(a, \text{extract\_choice}(o_i))) \\ 0, \text{otherwise.} \end{cases}$$

Following Qian et al. (2025), we omit the KL penalty term from the original GRPO objective to encourage more effective exploitation of reward signals during policy updates. Building on this, we design a verifiable reward system for training generative reward models in the tool-use scenario.

## 3 EXPERIMENTS

### 3.1 DO TOOLRM PROVIDE PRECISE REWARDS?

**Benchmark Construction.**    We evaluate the reward models using an improved benchmark adapted from IBM Research[4], based on BFCL. The original benchmark pairs correct function calls with incorrect ones generated by 25 permissively licensed models but has two main limitations: (1) it only covers single-turn tasks, and (2) its negative responses are too trivial for powerful RMs to differentiate. To overcome this, we construct a more challenging benchmark using the *multi_turn_base* split from BFCL V3 and curate harder negative samples from seven top-performing function-calling models[5]: xLAM-2-70B-FC-R (Prabhakar et al., 2025), GPT-4o (Hurst et al., 2024), OpenAI o1 (Jaech et al., 2024), Qwen3-32B (Yang et al., 2025), DeepSeek-R1 (Guo et al., 2025a), Gemini-2.5-Pro (Comanici et al., 2025), and Claude-3.7-Sonnet (Anthropic, 2025).

The resulting benchmark, TRBENCH$_{\text{BFCL}}$, comprises 2,983 samples from 1,397 unique tasks across 9 splits: simple (S), multiple (M), parallel (P), parallel multiple (PM), live sample (LS), live multiple (LM), live parallel (LP), live parallel multiple (LPM), and multi-turn base (MTB). It covers 20 distinct error types with rejected responses from 7 different models. Since BFCL tasks and their synthetic data are excluded from training, **TRBENCH$_{\text{BFCL}}$ serves as a strong OOD evaluation set for TOOLRM**. Additional statistics and implementation details are in Appendix C.2.

**Evaluation Metric.**    We assess reward model performance via pairwise preference classification. To minimize position bias, each sample is evaluated twice, swapping the response order on the second pass. A sample is correct only if both orders yield the correct prediction. For scalar-output RMs, we compute scores for chosen and rejected responses and mark the result correct if the score order matches the preference label. We report average accuracy (**Avg.**) across splits and weighted-average accuracy (**W-Avg.**), based on sample counts.

**Model Training.**    We train reward models on three reasoning-capable models (Qwen3-4B, Qwen3-8B, and Qwen3-4B-Thinking-2507) and four non-reasoning models (Qwen3-4B-Instruct-2507, Llama-3.2-3B-Instruct, Llama-3.1-8B-Instruct (Dubey et al., 2024), and Llama-xLAM-2-8B-FC-R (Prabhakar et al., 2025)) across different model families. At both training and inference, we apply the appropriate *think-mode* or *no-think-mode* templates. Our preference dataset, *ToolPref-Pairwise-30K*, contains 30,000 samples (29,500 for training, 500 for validation), built with our proposed pipeline. See training details in Appendix C.1 and impact of data scaling on ToolRM in Appendix D.

**Baseline Models.**    We benchmark ToolRM on TRBENCH$_{\text{BFCL}}$ against strong LLMs in the LLM-as-a-judge setup, including GPT, Gemini, Claude, DeepSeek, and Qwen. Specialized models are also tested: generative (Skywork-Critic (Shiwen et al., 2024), M-Prometheus (Pombal et al., 2025), RM-R1 (Chen et al., 2025c), RRM (Guo et al., 2025b)), discriminative (Skywork-Reward (Liu et al., 2024a), InternLM2-Reward (Cai et al., 2024)), and hybrid (Cloud-RM (Ankner et al., 2024)).

**Main Results.**    Table 2 presents evaluation results on TRBENCH$_{\text{BFCL}}$ across all splits. Training on *ToolPref-Pairwise-30K* significantly boosts performance, yielding an average gain of 10.12% and a maximum of 14.28% in weighted accuracy. ToolRM, trained on Qwen3-4B-Thinking-2507, consistently outperforms nearly all baselines, including on the *multi-turn-base* split—despite being

---

[4]https://huggingface.co/datasets/ibm-research/fc-reward-bench
[5]Trajectories from https://github.com/HuanzhiMao/BFCL-Result

Table 2: Evaluation results of reward models on TRBench$_{\text{BFCL}}$. A higher percentage of accuracy indicates a stronger ability to distinguish the better response in tool-calling tasks. The best result in each column is **bolded**, and the second-best is underlined. (◇): evaluated with the *think-mode* template; (♡): evaluated with the *no-think-mode* template; (♣): evaluated with the official template. (⚖): pairwise inputs; (◎): pointwise inputs; (💬): critique as output; (✅): choice as output; (🔢): scalar reward as output. Models trained in this paper are indicated with a green background .

| Models | Classification Accuracy (%) | | | | | | | | | | |
|---|---|---|---|---|---|---|---|---|---|---|---|
| | S | M | P | PM | LS | LM | LP | LPM | MTB | **Avg.** | **W-Avg.** |
| *Proprietary & Open-source Frontier LLMs* | | | | | | | | | | | |
| DeepSeek-AI/DeepSeek-R1-0528◇ | 68.61 | 70.42 | 87.71 | 85.64 | 64.62 | 46.45 | 76.47 | 75.00 | 36.77 | 67.97 | 57.93 |
| OpenAI/GPT-4o-2024-11-20♡ | 69.34 | 66.20 | 86.71 | 86.67 | 50.47 | 50.82 | 67.65 | 78.33 | 38.38 | 66.06 | 59.00 |
| OpenAI/o3-2025-04-16◇ | 70.80 | 69.01 | 85.71 | 84.87 | 55.19 | 50.43 | 67.65 | 76.67 | 41.21 | 66.84 | 59.40 |
| Google/Gemini-2.5-Flash◇ | 64.23 | 66.20 | 89.70 | 89.49 | 56.13 | 51.13 | 79.41 | 80.00 | 36.77 | 68.12 | 59.87 |
| Google/Gemini-2.5-Pro◇ | 75.18 | 67.61 | 88.04 | 91.79 | 58.96 | 48.32 | 82.35 | 73.33 | 39.80 | 69.49 | 59.94 |
| Qwen/Qwen3-235B-A22B-Thinking-2507◇ | 71.53 | 69.01 | 86.05 | 90.26 | 67.92 | 51.52 | 85.29 | 76.67 | 34.55 | 70.31 | 60.64 |
| DeepSeek-AI/DeepSeek-V3-0324♡ | 75.18 | 66.20 | 88.70 | 89.74 | 58.02 | 53.86 | 70.59 | 73.33 | 37.17 | 68.09 | 61.45 |
| Qwen/Qwen2.5-Max♡ | 77.37 | 73.24 | 89.04 | 90.00 | 58.02 | 55.18 | 67.65 | 70.00 | 37.98 | 68.72 | 62.39 |
| Anthropic/Claude-3.7-Sonnet◇ | 76.64 | 67.61 | 91.69 | **92.82** | 60.85 | 52.77 | 73.53 | 78.33 | 39.19 | 70.38 | 62.45 |
| Anthropic/Claude-4-Sonnet◇ | 81.02 | 77.46 | 91.36 | 91.28 | 62.74 | 54.95 | 82.35 | 83.33 | 41.01 | 73.95 | 64.23 |
| *Open-source Reward Models* | | | | | | | | | | | |
| Databricks/CLoud-RM-Llama-3-8B♣ | 25.55 | 35.21 | 33.22 | 32.82 | 31.60 | 37.88 | 32.35 | 25.00 | 49.90 | 33.73 | 37.34 |
| Unbabel/M-Prometheus-7B♡ | 54.74 | 54.93 | 71.43 | 74.87 | 43.87 | 46.69 | 38.24 | 53.33 | 34.14 | 52.47 | 51.19 |
| Microsoft-Research/RRM-7B◇ | 65.69 | 56.34 | 82.06 | 84.62 | 43.40 | 49.65 | 44.12 | 68.33 | 36.36 | 58.95 | 56.05 |
| UIUC/RM-R1-DeepSeek-Distilled-Qwen-32B◇ | 75.18 | 76.06 | 68.44 | 80.51 | 61.79 | 49.18 | 52.94 | 53.33 | 38.18 | 61.73 | 56.25 |
| Unbabel/M-Prometheus-14B♡ | 64.96 | 57.75 | 88.37 | 87.44 | 44.34 | 46.38 | 64.71 | 61.67 | 39.39 | 61.67 | 56.32 |
| Skywork/Skywork-Critic-Llama-3.1-8B♣ | 54.74 | 59.15 | 86.05 | 83.59 | 47.17 | 45.75 | 67.65 | 61.67 | 50.30 | 61.79 | 56.92 |
| Skywork/Skywork-Critic-Llama-3.1-70B♣ | 64.23 | 67.61 | 87.38 | 88.21 | 44.34 | 46.69 | 70.59 | 66.67 | 47.47 | 65.35 | 60.31 |
| Microsoft-Research/RRM-32B◇ | 76.64 | 76.06 | 87.38 | 89.23 | 67.92 | 56.90 | 67.65 | 75.00 | 42.83 | 71.07 | 64.50 |
| Skywork/Skywork-Reward-Llama-3.1-8B-v0.2♣ | **83.21** | 70.42 | **92.36** | 92.31 | 59.91 | 62.51 | 67.65 | 75.00 | 59.80 | 73.68 | 70.23 |
| InternLM/InternLM2-7B-Reward♣ | 80.29 | **80.28** | 88.04 | 89.74 | 63.68 | 65.16 | 67.65 | 73.33 | **61.21** | 74.38 | 71.17 |
| *Models Trained on ToolPref-Pairwise-30K* | | | | | | | | | | | |
| Meta/Llama-3.2-3B-Instruct♡ | 34.31 | 33.80 | 24.58 | 34.87 | 26.89 | 29.54 | 8.82 | 30.00 | 20.20 | 27.00 | 28.09 |
| TOOLRM-Llama-3.2-3B-Instruct♡ | 54.01 | 57.75 | 87.04 | 78.97 | 44.34 | 54.95 | 64.71 | 61.67 | 45.45 | 60.99 | 59.27 (+ 31.18) |
| Meta/Llama-3.1-8B-Instruct♡ | 45.99 | 52.11 | 46.84 | 62.31 | 33.02 | 39.44 | 23.53 | 40.00 | 28.69 | 41.33 | 41.38 |
| TOOLRM-Llama-3.1-8B-Instruct♡ | 62.04 | 61.97 | 88.70 | 86.15 | 47.64 | 52.30 | 82.35 | 68.33 | 41.21 | 65.63 | 59.57 (+18.19) |
| Salesforce/Llama-xLAM-2-8B-FC-R♡ | 48.91 | 36.62 | 74.09 | 72.05 | 26.89 | 29.62 | 44.12 | 51.67 | 32.32 | 46.25 | 41.59 |
| TOOLRM-Llama-xLAM-2-8B-FC-R♡ | 51.09 | 54.93 | 63.12 | 55.64 | 41.98 | 51.52 | 67.65 | 55.00 | 40.40 | 53.48 | 51.02 (+9.43) |
| Qwen/Qwen3-4B-Instruct-2507♡ | 71.53 | 64.79 | 90.37 | 89.23 | 51.42 | 50.66 | 70.59 | 86.67 | 36.57 | 67.98 | 59.67 |
| TOOLRM-Qwen3-4B-Instruct-2507♡ | 70.80 | 74.65 | 91.03 | 89.49 | 55.66 | 60.41 | **94.12** | 81.67 | 49.90 | 74.19 | 66.85 (+7.18) |
| Qwen/Qwen3-4B (Thinking mode)◇ | 70.07 | 73.24 | 89.70 | 87.69 | 56.60 | 48.09 | 79.41 | 81.67 | 39.80 | 69.59 | 59.34 |
| TOOLRM-Qwen3-4B◇ | 81.02 | 78.87 | 89.04 | 88.97 | 63.21 | 62.12 | 91.18 | 86.67 | 52.32 | 77.04 | 68.89 (+9.55) |
| Qwen/Qwen3-8B (Thinking mode)◇ | 71.53 | 61.97 | 89.37 | 90.26 | 58.49 | 48.09 | 85.29 | 76.67 | 39.19 | 68.98 | 59.44 |
| TOOLRM-Qwen3-8B◇ | 81.02 | 76.06 | 89.70 | 91.03 | 64.62 | 61.50 | 91.18 | 80.00 | 52.73 | 76.43 | 68.92 (+9.48) |
| Qwen/Qwen3-4B-Thinking-2507◇ | 67.88 | 70.42 | 85.71 | 87.69 | 61.79 | 46.61 | 85.29 | 85.00 | 33.54 | 69.33 | 57.59 |
| **TOOLRM-Qwen3-4B-Thinking-2507◇** | **83.21** | **80.28** | 90.03 | 92.56 | **71.23** | **66.02** | **94.12** | **88.33** | 52.12 | **79.77** | **71.87** (+14.28) |

trained on step-wise critiques. Since BFCL scoring for multi-turn tasks relies on state- and response-based signals rather than rule-matching, these gains demonstrate that **ToolRM acquires robust, generalizable analytical capabilities rather than overfitting to rule-based labels.**

In LLM-as-a-judge evaluations, Claude models outperform other frontier LLMs, aligning with their tool-use strengths. Among specialized reward models, InternLM2-7B-Reward performs best, likely due to its diverse training on 2.4 million preference pairs spanning dialogue, code, and math. Interestingly, Skywork-Reward-Llama-3.1-8B-v0.2 surpasses its generative counterpart Skywork-Critic, despite both being trained on similar datasets. This suggests that, without targeted training, scalar-output discriminative RMs may generalize better to tool-use tasks than generative critics.

Lastly, reasoning (thinking) models show greater gains from critique training than instruction-tuned counterparts, and models with longer initial reasoning patterns (e.g., Qwen3-4B-Thinking-2507 vs. Qwen3-4B) benefit the most. This highlights that **even with weaker initial performance, a greater capacity for exploration can ultimately lead to stronger outcomes through RL.** A comparison between DeepSeek-R1 and DeepSeek-V3 further emphasizes the pivotal role of high-quality data in enhancing models' reasoning abilities on targeted tasks.

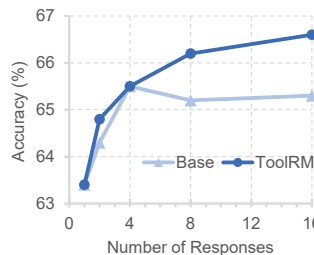

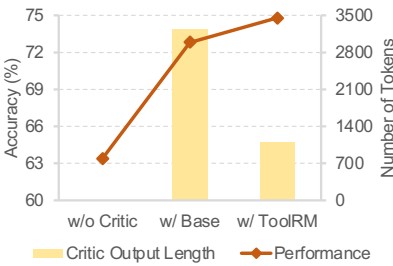

Figure 2: Comparison of BoN sampling on ACEBench.

Figure 3: Comparison of model self-correction on ACEBench.

### 3.2 Do ToolRM Help with Inference-Time Scaling?

**Setup.** We assess whether ToolRM improves tool-call inference using 823 samples from the *Normal* split of ACEBENCH (Chen et al., 2025a), a benchmark for tool-use evaluation. For each sample, we apply Best-of-N (BoN) sampling with Qwen3-4B-Instruct-2507 (temperature = 1.0), and use generative reward models to select the best response. We compare two judges: the baseline Qwen3-4B-Thinking-2507 (*Base*) and our trained ToolRM-Qwen3-4B-Thinking-2507 (*ToolRM*). Performance is measured by average accuracy across all samples.

**Main Results.** Figure 2 shows that ToolRM consistently matches or outperforms the baseline across all BoN settings, with gains of 3.2 and 1.3 points over the non-BoN and BoN-16 baselines, respectively. These improvements suggest that RL training enhances underlying reasoning, enabling effective generalization beyond the original training context. Notably, ToolRM maintains stable performance as the candidate pool grows, demonstrating its **robustness to long-context reasoning** and its **utility in inference-time scaling for tool-use tasks.**

### 3.3 Do Critiques Improve Model Self-Correction?

**Setup.** We assess the effectiveness of critiques generated by ToolRM in guiding policy model self-correction (SC). For each sample in the *Normal* subset of ACEBench, Qwen3-4B-Instruct-2507 first produces a function-calling response. A generative reward model then critiques this output with concise feedback. Using this critique, the same model edits its response. We compare two critics: the baseline Qwen3-4B-Thinking-2507 (*Base*) and our trained ToolRM-Qwen3-4B-Thinking-2507 (*ToolRM*). Performance is measured by average accuracy over all samples.

**Main Results.** As shown in Figure 3, ToolRM leads to notable gains in self-correction accuracy: +11.4 points over *w/o Critic* and +2.0 over *w/ Base*, confirming its ability to produce reliable, targeted critiques. Additionally, ToolRM achieves this with much lower decoding cost—reducing average output length from 3,211 to 1,111 tokens—demonstrating **efficient reasoning without sacrificing critique quality.** See Appendix H for more qualitative examples.

### 3.4 Ablation Studies on Preference Data Construction

To assess the contribution of our two key data construction components, we conduct an ablation study with two sets of variants. In the first set, we replace balanced multi-dimensional sampling with random sampling (*w/o BMDS*) and perform fine-grained ablations along three critical dimensions: diversity of data sources (*w/o DDS*), coverage of preference intensity (*w/o CPI*), and complexity of tasks (*w/o CT*). In the second set, we remove the unified evaluation criteria during training (*w/o EC*). Models are trained using GRPO on Qwen3-4B-Thinking-2507 with 30K pairwise preferences, keeping all other settings fixed. As shown in Table 3, removing either component sig-

Table 3: Ablated evaluation results on TRBench$_{\text{BFCL}}$.

| Model | W-Avg. Acc |
|---|---|
| Full ToolRM | 71.87 |
| - *w/o Full BMDS* | 67.24 (-4.63) |
| - *w/o DDS* | 68.64 (-3.23) |
| - *w/o CPI* | 70.29 (-1.58) |
| - *w/o CT* | 68.89 (-2.98) |
| - *w/o EC* | 68.69 (-3.18) |

nificantly degrades performance. Each BMDS dimension contributes to performance; diversity of data sources and task complexity have larger effects than preference intensity, underscoring the importance of both diversity and contextual complexity for reward-model training. Moreover, output

length of models decreases sharply without the evaluation criteria (1,204→694), suggesting these criteria promote more comprehensive reasoning during training.

## 3.5 Ablation Studies on Model Training

**Data Domain.** We investigate the influence of in-domain preference data on reward model performance by conduct the following experiments: (i) we randomly sample 30,000 instances from *Skywork-Reward-Preference-80K-v0.2* (Liu et al., 2024a), a high-quality general preference dataset; (ii) we make minimal modifications to ToolRM prompt template (removing the original evaluation criteria) and use it to perform RL training on the baseline models in the same way as for previous ToolRM; (iii) the trained models are then evaluated on TRBench$_{BFCL}$ where evaluation results are labeled with *NormalPref* in Table 4. According to the results, models trained on high-quality normal preference data do improve their judging performance on pairwise classification

Table 4: Evaluation results of different variants on TRBench$_{BFCL}$.

| Model | W-Avg. Acc |
|---|---|
| Qwen3-4B-Instruct-2507 | 59.67 |
| - GenRM on *NormalPref* | 63.82 (+4.15) |
| **- GenRM on *ToolPref*** | **66.85 (+7.18)** |
| - ScalarRM on *NormalPref* | 67.88 (+8.21) |
| **- ScalarRM on *ToolPref*** | **77.61 (+17.94)** |
| Qwen3-4B-Thinking-2507 | 57.59 |
| - GenRM on *NormalPref* | 63.19 (+5.60) |
| **- GenRM on *ToolPref*** | **71.87 (+14.28)** |
| - ScalarRM on *NormalPref* | 69.69 (+12.10) |
| **- ScalarRM on *ToolPref*** | **76.80 (+19.21)** |

tasks in the tool-use domain. However, the in-domain preference dataset delivers substantially larger gains over base models, particularly when training from a think-version base model.

**Training Objective.** Following Liu et al. (2024a), we further investigate the impact of ToolPref-Pairwise-30K on training discriminative reward models (ScalarRM) using the Bradley–Terry (BT) objective (Bradley & Terry, 1952). As shown in Table 4, the constructed dataset remains effective under BT objective and can further improve RM performance compared with the RL objective in pairwise preference classification tasks. This is consistent with our previous findings on the Skywork-Critic/Reward model series:

Table 5: Evaluation results of different variants on ACEBench.

| Model | Acc |
|---|---|
| Qwen3-4B-Instruct-2507 | 63.4 |
| - BoN-16 w/ GenToolRM | 66.6 (+3.2) |
| - BoN-16 w/ ScalarToolRM | 67.2 (+3.8) |
| **- SC w/ GenToolRM** | **74.8 (+11.4)** |

when using the same base model and training data, ScalarRM trained with a BT objective naturally produces more accurate relative scores than GenRM. We also observe that instruct-tuned base models, which produce more concise outputs, are better suited to train ScalarRM with a BT objective for generating precise scores, whereas think-version models, which produce longer initial chain of thoughts and exhibit stronger exploration capability, are better suited for RL training to obtain GenRM with stronger analytical ability. As shown in Table 5, ScalarToolRM yields larger gains when used to judge best-of-N sampling, whereas GenToolRM is substantially more effective when used to provide self-correction feedback. In practice, each training objective has distinct strengths and should be chosen according to the application scenario: use GenRM when critique-style feedback and interpretability are required (e.g., self-correction), and use ScalarRM when only accurate reward scoring is needed (e.g., RL training or BoN sampling).

## 3.6 Error Analysis of ToolRM

We further analyze the thinking process of ToolRM in cases where its final judgments are inconsistent with the ground-truth preferences. Our investigation indicates that these errors primarily fall into two categories: (i) when the description of tool schema or parameters lacks concrete examples, the model is unable to infer the most appropriate tool invocation from the candidates, given the available tool information and the user's query; (ii) the originally annotated chosen response contains minor errors, while the rejected response has more fundamental and severe errors. The model correctly identifies all errors but fails to distinguish primary errors from secondary ones, leading to an incorrect pairwise reward. We believe the first type of error is constrained by the base model's inherent reasoning capability and is therefore more difficult to improve. The second type, however, is more tractable and can be mitigated through targeted optimization using higher-quality, non-perfect preference pairs, of which the chosen response still contains minor errors. Examples from TRBench$_{BFCL}$ corresponding to the two typical error types are provided in Appendix H for detailed reference.

## 4 RELATED WORK

**Tool Learning in the Era of LLMs.** Early work on agentic AI, such as Yao et al. (2023), combines chain-of-thought reasoning (Wei et al., 2022) with tool-augmented actions to elicit LLMs' tool-use capabilities. Later methods imitate curated tool-use trajectories via supervised fine-tuning (Schick et al., 2023; Liu et al., 2024b), but often struggle with complex or out-of-distribution tasks. More recently, researchers have integrated verified rewards into tool-aware reasoning, with designs tailored for question answering (Jin et al., 2025; Song et al., 2025), math (Feng et al., 2025; Dong et al., 2025), and general tool-use (Qian et al., 2025; Zhang et al., 2025).

**Reward Modeling.** Reward models guide large language models toward outputs that align with human preferences (Ouyang et al., 2022; Bai et al., 2022). They are typically either (1) discriminative, outputting scalar scores to rank responses (Cai et al., 2024; Liu et al., 2024a), or (2) generative, producing textual rewards for domains such as chat (Shiwen et al., 2024), code (McAleese et al., 2024), and literary translation (Pombal et al., 2025). A recent trend views reward modeling as a reasoning process (Chen et al., 2025c; Guo et al., 2025b) to enhance reward quality. Following this line of work, we extend generative reward modeling to the field of tool calling in this paper.

## 5 CONCLUSION

This paper presents TOOLRM, a family of generative reward models tailored for agentic tool-use tasks. Central to our framework is a novel data construction pipeline that combines rule-based labeling with balanced multi-dimensional sampling. This approach enables the automatic generation of fine-grained pairwise preference data, yielding a dataset that is diverse, well-balanced, and deliberately challenging. The resulting dataset supports efficient RL-based training and encourages the development of nuanced reasoning strategies beyond surface-level signal matching.

By formulating the reward modeling objective as a discriminative critique task, and optimizing via RLVR, TOOLRM not only learns to assign scalar preferences but also acquires robust and generalizable analytical capabilities. Our comprehensive evaluation across multiple benchmarks confirms the utility of TOOLRM in three key dimensions: (i) delivering high-fidelity reward signals that align with human preferences and outperform frontier baselines; (ii) enabling inference-time scaling by reliably selecting optimal outputs from diverse candidate pools; and (iii) providing efficient and effective pointwise critiques that improve self-correction with minimal decoding overhead.

These results collectively suggest that reward models, when trained on structured critique data, can evolve into capable reasoning agents, capable of supporting downstream decision-making in real-world LLM applications. Future work may explore extending this approach to more open-ended agentic tasks, incorporating human-in-the-loop feedback, and leveraging generative critics to guide multi-agent coordination and planning.

## REPRODUCIBILITY STATEMENT

To promote reproducibility, prompt templates for model training and inference across all experiments are shown in Appendix G. All open-source models used in our experiments are obtained from their official HuggingFace repositories[6]. In addition to the main text, Appendix C offers further implementation details on benchmark construction and the experimental setup. To facilitate reproduction of the proposed data sampling strategy *BMDS*, we include a detailed description and pseudocode in Appendix E. We will open-source the trained reward-model series TOOLRM, together with the training dataset *ToolPref-Pairwise-30K*, and the enhanced benchmark TRBench$_{\text{BFCL}}$ to advance future research in this field.

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

## A  THE USE OF LARGE LANGUAGE MODELS

During the completion of this work, we employed Gemini 2.5 Pro (Comanici et al., 2025) to identify grammatical errors and refine the text in the preliminary draft stage. The data construction pipeline code was initially developed by the human authors and then verified using Qwen3-Coder (Yang et al., 2025). All suggestions from the LLMs were manually reviewed and confirmed for accuracy.

## B  FULL RELATED WORK

### B.1  TOOL LEARNING IN THE ERA OF LLMS

The emergence of foundational capabilities in large language models (LLMs) has enabled them to identify and use appropriate tools in a human-like manner. Yao et al. (2023) unlock this ability by combining chain-of-thought reasoning (Wei et al., 2022) with tool-augmented actions. Another line of approaches clones behaviors from completed tool-calling trajectories using supervised fine-tuning (Schick et al., 2023; Tang et al., 2023; Liu et al., 2024b; 2025a), while these methods may face challenges generalizing to complex and out-of-distribution tasks. To address this limitation, other approaches employ reinforcement learning with human preference data to learn via trial-and-error (Nakano et al., 2021). Building on recent successes in reasoning models (Lambert et al., 2025; Shao et al., 2024), utilizing verified rewards to facilitate tool-integrated reasoning has become a promising direction. Reward designs based on the format and correctness of the final answer have proven effective in tasks like question-answering (Jin et al., 2025; Song et al., 2025), math (Feng et al., 2025; Dong et al., 2025), and general tool-calling (Qian et al., 2025; Zhang et al., 2025), leading to generalized model improvements through reinforcement learning.

### B.2  EVALUATION OF LLM TOOL-USE

Numerous tool-calling benchmarks have been proposed in recent years. To enable realistic and reliable evaluation, tasks are either drawn from real-world domains (Wang et al., 2024; Patil et al., 2024; Zhong et al., 2025; Yao et al., 2025; Barres et al., 2025) or generated via well-designed data-synthesis pipelines (Qin et al., 2024; Chen et al., 2025a). Among these, BFCL (Patil et al., 2025) covers diverse and complex patterns of tool usage and serves as a comprehensive benchmark for evaluating LLMs' tool-use capabilities. Nevertheless, there remains a lack of a benchmark that assesses whether current models can provide accurate feedback on LLM actions in tool-use scenarios.

### B.3  REWARD MODELING OF HUMAN PREFERENCES

Reinforcement learning has proven effective for aligning LLMs with human preferences, using feedback from humans (Ouyang et al., 2022) or other capable LLMs (Bai et al., 2022; Lee et al., 2024). Central to this process are reward models (RMs), which are primarily developed in two ways. The first is discriminative modeling, where RMs output a scalar score to differentiate between preferred and rejected responses (Yang et al., 2024; Cai et al., 2024; Liu et al., 2024a; 2025b). The second is generative modeling, where models provide textual rewards as natural language critiques for tasks like chat (Shiwen et al., 2024; Kim et al., 2024; Yu et al., 2025), code (McAleese et al., 2024), and literary machine translation (Pombal et al., 2025). Hybrid approaches combine critiques with scalar rewards to better capture nuanced preferences (Ankner et al., 2024; Wang et al., 2025), while recent work frames reward modeling as reasoning tasks (Chen et al., 2025c; Wang et al., 2025; Guo et al., 2025b; Whitehouse et al., 2025). In this paper, we extend generative reward modeling to general tool use, offering textual critiques as valuable feedback.

Notably, there is also a line of work on tool-augmented reward modeling Li et al. (2024); Findeis et al. (2025); Xu et al. (2025), which is conceptually distinct from ToolRM, with different motivations and inference procedures. In our setting, ToolRM is trained to evaluate another policy model's behavior on agentic tool-use tasks, and it relies solely on internal reasoning rather than invoking external tools during evaluation. By contrast, tool-augmented RMs are primarily designed for target tasks such as general QA, writing, and coding, where the policy model can complete the task without invoking any tools, and tools are instead called at evaluation time to improve the reliability of

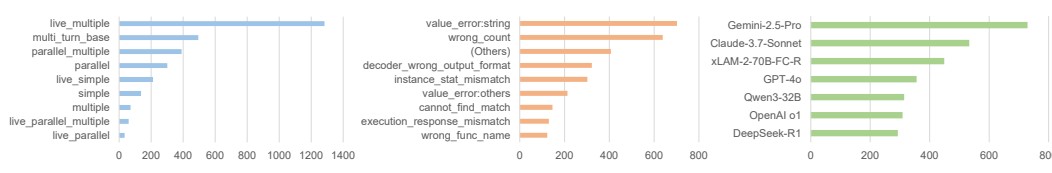

(a) Distribution of data patterns.    (b) Distribution of error types.    (c) Distribution of response sources.

Figure 4: Statistics of the enhanced reward model benchmark TRBench$_{\text{BFCL}}$.

reward estimates. Consequently, these RMs do not apply to the scenario studied in this paper and are not directly comparable to our approach.

## C EXPERIMENT DETAILS

### C.1 MODEL TRAINING

We train the reward models on eight NVIDIA A100 80G GPUs. We perform one epoch of GRPO training using veRL (Sheng et al., 2025), with a learning rate of `1e-6` and a clip ratio of $\epsilon = 0.2$. At each training step, we sample a batch of 128 queries and generate 8 trajectories per query. Trajectory generation is handled by the vllm backend (Kwon et al., 2023), employing sampling hyper-parameters of `temperature=1.0`, `top_p=1.0`, and `top_k=-1`. Due to resource constraints, we limit the maximum prompt length to 16,384 tokens and the maximum response length to 4,096 tokens for model training.

### C.2 BENCHMARK IMPLEMENTATION

In constructing TRBENCH$_{\text{BFCL}}$, we prepare preference pairs for each data task according to its turn-wise trajectory pattern. For single-turn tasks (splits originally introduced in BFCL v1 and v2), evaluation is based on the Abstract Syntax Tree (AST), which compares a model-generated function against its function documentation and a set of possible correct answers. In these cases, we source the oracle answers directly from the benchmark as the *chosen* responses and extract incorrect responses from the failed trajectories, forming *chosen–rejected* pairs for each task.

For multi-turn tasks (the split introduced in BFCL v3), evaluation instead relies on state-based and response-based checks[7], which differ from the rule-based matching used to check tool calls in building $\mathcal{D}_{\text{pref}}$. In these complex scenarios, while pinpointing the single failing tool call is difficult, one can easily identify the entire incorrect turn by comparing the generated trajectory to the ground truth. We leverage this to create evaluation pairs: the incorrect output is the concatenation of all tool calls the model generated in that turn, and the correct output is the concatenation of all tool calls from the corresponding ground-truth solution. We show statistics of the enhanced reward model benchmark TRBench$_{\text{BFCL}}$ in Figure 4.

To ensure fair evaluation across different types of baseline models, we first apply the same *think-mode/no-think-mode* template used in our model evaluations. If the test model is unable to follow the specific instruction, we instead evaluate it using its official prompt. To fully harness the potential of the test models, the official default sampling parameters are used for inference, except that the maximum output length is limited to 8,192 tokens to prevent excessively long and repetitive chain-of-thought content.

## D IMPACT OF DATA SCALING ON TOOLRM

We investigate the influence of data scaling on model performance. Figure 5a shows the results for Qwen3-4B-Thinking-2507 on TRBench$_{\text{BFCL}}$, trained with data samples ranging from 10K to 40K. Notably, the model achieves its highest performance with 30K training samples. Performance does not increase monotonically with data size because our sampling strategy prioritizes more complex

---

[7]https://gorilla.cs.berkeley.edu/blogs/13_bfcl_v3_multi_turn.html

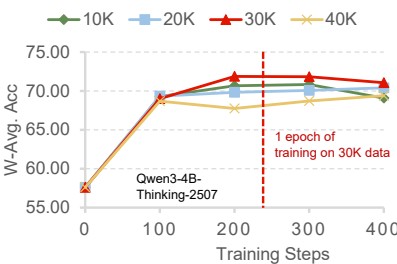 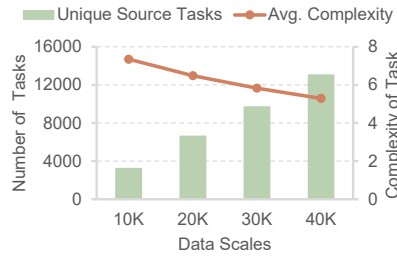

(a) Evaluation results on TRBench_BFCL.  (b) Statistics across different data scales.

Figure 5: Statistics and the impact of data scale on model training.

tasks. As the dataset grows, the average task complexity declines, leading to less effective training signals. Figure 5b illustrates this trend: while the number of unique tasks rises with larger datasets, their average complexity decreases. These results demonstrate that our proposed strategy successfully balances task diversity and complexity when exploring the candidate data pool.

# E   THE BALANCED MULTI-DIMENSIONAL SAMPLING ALGORITHM

In this section, we detail the implementation of the BMDS strategy for efficient sampling. To discretize the distribution of preference intensities $I_{\text{preference}}$ among data samples, we initialize a set of bins $B = \{b_0, b_1, \ldots, b_m\}$ with fixed intervals. In our experiments, we set: $B = \{(0, 0.1], (0.1, 0.2], \ldots, (0.9, 1]\}$. Each sample in the candidate pairwise data pool $\mathcal{D}_{\text{pair-cand}}$ is assigned to the corresponding bin, indexed from 0 to $m$, according to its preference intensity. We then group the samples by a composite key (`source`, `bin_index`) to ensure representation across different data sources and varying preference intensities. Within each group, samples are sorted in descending order of task complexity $S_{\text{complexity}}$. Sampling proceeds greedily: we first exhaustively select all samples from the group with the fewest entries, and then allocate the remaining quota as evenly as possible across the other bins. This yields a diverse, well-balanced, and sufficiently challenging subset of data. We present pseudocode of this strategy in Algorithm 1.

---

**Algorithm 1** Balanced Multi-Dimensional Sampling Strategy

---

**Iutput:** Data pool $\mathcal{D}_{\text{pair-cand}}$, bin edges $B$, target sample size $N$
**Output:** A subset $\mathcal{D}_{\text{pair-sampled}}$ of diverse, balanced, and challenging samples
1: *# Step 0: Check data sufficiency*
2: **if** $|\mathcal{D}_{\text{pair-cand}}| < N$ **then**
3:    **raise** InsufficientDataError
4: **end if**
5: *# Step 1: Assign samples to bins*
6: **for** each $d_i \in \mathcal{D}_{\text{pair-cand}}$ **do**
7:    $d_i.\texttt{bin\_idx} \leftarrow \text{assign}(d_i.I_{\text{preference}}, B)$
8: **end for**
9: *# Step 2: Group by composite key*
10: Initialize group dictionary $\mathcal{G} \leftarrow \emptyset$
11: **for** each $d_i \in \mathcal{D}_{\text{pair-cand}}$ **do**
12:    $key \leftarrow (d_i.\texttt{source}, d_i.\texttt{bin\_idx})$
13:    $\mathcal{G}[key] \leftarrow \mathcal{G}[key] \cup \{d_i\}$
14: **end for**
15: *# Step 3: Sort within each group by task complexity (descending)*
16: **for** each group $G \in \mathcal{G}$ **do**
17:    $G \leftarrow \text{sort}(G, \text{key} = S_{\text{complexity}}, \text{order=descending})$
18: **end for**
19: *# Step 4: Sort groups by size (ascending)*
20: $\mathcal{G}_{\text{sorted}} \leftarrow \text{sort}(\mathcal{G}.\text{values}(), \text{key} = |G|, \text{order=ascending})$
21: *# Step 5: Greedy allocation*
22: Initialize sampling quotas: $Q \leftarrow [0] \times |\mathcal{G}_{\text{sorted}}|$
23: $N_{\text{remaining}} \leftarrow N, k \leftarrow 0$
24: **while** $k < |\mathcal{G}_{\text{sorted}}|$ **and** $N_{\text{remaining}} > 0$ **do**
25:    $m \leftarrow |\mathcal{G}_{\text{sorted}}| - k$
26:    $n_{\text{avg}} \leftarrow \lceil N_{\text{remaining}}/m \rceil$
27:    **if** $|\mathcal{G}_{\text{sorted}}[k]| \leq n_{\text{avg}}$ **then**
28:      $Q[k] \leftarrow |\mathcal{G}_{\text{sorted}}[k]|$
29:      $N_{\text{remaining}} \leftarrow N_{\text{remaining}} - |\mathcal{G}_{\text{sorted}}[k]|$
30:      $k \leftarrow k + 1$
31:    **else**
32:      *# Distribute remaining quota evenly*
33:      $q \leftarrow \lfloor N_{\text{remaining}}/m \rfloor$
34:      $r \leftarrow N_{\text{remaining}} \mod m$
35:      **for** $i = k$ **to** $|\mathcal{G}_{\text{sorted}}| - 1$ **do**
36:        $Q[i] \leftarrow q$
37:      **end for**
38:      **for** $i = 0$ **to** $r - 1$ **do**
39:        $Q[|\mathcal{G}_{\text{sorted}}| - 1 - i] \leftarrow Q[|\mathcal{G}_{\text{sorted}}| - 1 - i] + 1$
40:      **end for**
41:      **break**
42:    **end if**
43: **end while**
44: *# Step 6: Sample data based on quotas*
45: $\mathcal{D}_{\text{pair-sampled}} \leftarrow \emptyset$
46: **for** $i = 0$ **to** $|\mathcal{G}_{\text{sorted}}| - 1$ **do**
47:    $\mathcal{D}_{\text{pair-sampled}} \leftarrow \mathcal{D}_{\text{pair-sampled}} \cup \mathcal{G}_{\text{sorted}}[i][: Q[i]]$
48: **end for**
49: **return** $\mathcal{D}_{\text{pair-sampled}}$

---

# F  EXAMPLE OF TOOL-USE TASK TRAJECTORY

During conversation order validation, we retain only trajectories that satisfy the following message-role transition rules: [system→user, user→assistant, assistant→user/tool,

tool→assistant]. In this work, tool responses are set into `user` messages for compatibility. Figure 6 shows a format-aligned example from BUTTON illustrating a tool-use task trajectory.

```
A Tool-Call Trajectory from BUTTON

{
    "role": "system",
    "content": "# Tools

    You may call one or more functions to assist with the user query.
    You are provided with function signatures within <tools></tools> XML tags:
    <tools>
    {"type": "function", "function": {"name": "get_items_by_color", "description": "Retrieve all items in the inventory filtered by a specified color.",
    "parameters": {"required": ["color", "date"], "type": "object", "properties": {"color": {"type": "string", "description": "The color of the items to be
    retrieved."}, "date": {"type": "string", "description": "The date for which the inventory is being checked, in YYYY-MM-DD format."}}}}}
    {"type": "function", "function": {"name": "count_items", "description": "Count the number of items in a given list.", "parameters": {"required":
    ["items"], "type": "object", "properties": {"items": {"type": "array", "description": "The list of items to be counted."}}}}}
    </tools>

    For each function call, return a json object with function name and arguments within <tool_call></tool_call> XML tags:
    <tool_call>
    {"name": <function-name>, "arguments": <args-json-object>}
    </tool_call>"
},
{
    "role": "user",
    "content": "Hi, I need to count the red and blue items in the warehouse inventory today and compare their quantities. The warehouse inventory data
    should have information about item colors and quantities. Can you help me with this?"
},
{
    "role": "assistant",
    "content": "<think>
    The user wants to count the red and blue items in the warehouse inventory today and compare their quantities. To accomplish this task, we need to:
    1. Retrieve the total number of red items in the warehouse inventory today.
    2. Retrieve the total number of blue items in the warehouse inventory today.
    3. Compare the counts of red and blue items.
    4. Provide the final answer based on the comparison.
    We will start by retrieving and counting the red items first.
    </think>
    <tool_call>
    {"name": "get_items_by_color", "arguments": {"color": "red", "date": "2023-10-05"}}
    </tool_call>"
},
{
    "role": "user",
    "content": "<tool_response>
    {"name": "get_items_by_color", "arguments": {"color": "red", "date": "2023-10-05"}, "results": {"items": [{"id": "001", "name": "Red T-shirt",
    "color": "red", "quantity": 50, "date_added": "2023-09-20"}, {"id": "002", "name": "Red Mug", "color": "red", "quantity": 120, "date_added": "2023-
    10-01"}, {"id": "003", "name": "Red Hat", "color": "red", "quantity": 30, "date_added": "2023-09-29"}]}}
    </tool_response>"
},
{
    "role": "assistant",
    "content": "<think>
    The function `get_items_by_color` returned a list of red items along with their quantities for the date 2023-10-05. The items are:
    - Red T-shirt: Quantity 50
    - Red Mug: Quantity 120
    - Red Hat: Quantity 30 We need to count the total quantity of these red items. We'll sum up the quantities of all the red items returned.
    </think>
    <tool_call>
    {"name": "count_items", "arguments": {"items": [{"id": "001", "name": "Red T-shirt", "color": "red", "quantity": 50, "date_added": "2023-09-20"},
    {"id": "002", "name": "Red Mug", "color": "red", "quantity": 120, "date_added": "2023-10-01"}, {"id": "003", "name": "Red Hat", "color": "red",
    "quantity": 30, "date_added": "2023-09-29"}]}}
    </tool_call>"
},
{
    "role": "user",
    "content": "<tool_response>
    {"name": "count_items", "arguments": {"items": [{"id": "001", "name": "Red T-shirt", "color": "red", "quantity": 50, "date_added": "2023-09-20"},
    {"id": "002", "name": "Red Mug", "color": "red", "quantity": 120, "date_added": "2023-10-01"}, {"id": "003", "name": "Red Hat", "color": "red",
    "quantity": 30, "date_added": "2023-09-29"}]}, "results": {"count": 3}}
    </tool_response>"
},
{
    "role": "assistant",
    "content": "<think>
    The `count_items` function returned a count of 3 for the red items. However, the task requires summing the quantities of the items, not just counting
    the number of different items. We need to manually sum the quantities of the red items, as the `count_items` function didn't provide the total quantity
    but rather the count of item types. The quantities are 50, 120, and 30. The total quantity of red items is $(50 + 120 + 30 = 200)$.
    Next, we should retrieve and count the blue items in the warehouse inventory for today.
    </think>
    <tool_call>
    {"name": "get_items_by_color", "arguments": {"color": "blue", "date": "2023-10-05"}}
    </tool_call>"
}
```

Figure 6: A format-aligned tool-use trajectory from BUTTON.

## G  PROMPTING TEMPLATES

We present the evaluator prompt templates for the pairwise critique task used in both training and inference. The *think-mode* and *no-think-mode* templates are shown in Figures 7 and 8, respectively. Figure 9 shows the prompt template used by the judge for the BoN sampling task, in which the $N$ sampled responses are inserted and labeled from 1 to $N$. Figures 10 and 11 show the prompt templates used by critic and editor for the self-correction task. Figure 12 presents the template of the system prompt in each tool-use trajectory.

---

**Serve as an Evaluator for Pairwise Critique Task (Think Mode)**

<task>
You are an expert evaluator of AI assistant performance. Given a complete user-assistant conversation history and two generated assistant responses, you are to conduct a thorough, fact-based, and comprehensive comparison. Based on specific evidence from your evaluation, make a clear choice of which response is superior. There may be a list of tools available to the assisatnt. The assistant starts with one or more cycles of (thinking about which tool to use -> performing tool call -> waiting for tool response), and ends with (thinking about the answer -> answer of the question). The thinking processes, tool calls, tool responses, and answer are enclosed within their tags. There could be multiple thinking processes, tool calls, tool call parameters and tool response parameters.
</task>

<evaluation_criteria>
- Available tools must be fully and appropriately leveraged to meet the requirements.
- Tool call names must be valid, correct, and complete.
- Tool call arguments must be valid, correct, and complete.
- Fabrication, including the creation of information or knowledge not provided by the user, conflicting with user input, or not derived from the tools, must be penalized.
- Repetitive or unnecessary tool calls must be penalized.
- Excessive or unnecessary requests for user clarification beyond what is essential must be penalized.
</evaluation_criteria>

<conversation_history>
{chat_history}
</conversation_history>

<current_response_1>
{assistant_response_1}
</current_response_1>

<current_response_2>
{assistant_response_2}
</current_response_2>

Output your choice (either '1' or '2') within <choice></choice> XML tags. No explanations should precede or follow the choice. Answer in the following format.
<choice>
{{your_choice}}
</choice>

---

Figure 7: Evaluator prompt template of the pairwise critique task for reasoning LLMs.

---

**Serve as an Evaluator for Pairwise Critique Task (No Think Mode)**

<task>
You are an expert evaluator of AI assistant performance. Given a complete user-assistant conversation history and two generated assistant responses, you are to conduct a thorough, fact-based, and comprehensive comparison. Based on specific evidence from your evaluation, make a clear choice of which response is superior. There may be a list of tools available to the assisatnt. The assistant starts with one or more cycles of (thinking about which tool to use -> performing tool call -> waiting for tool response), and ends with (thinking about the answer -> answer of the question). The thinking processes, tool calls, tool responses, and answer are enclosed within their tags. There could be multiple thinking processes, tool calls, tool call parameters and tool response parameters.
</task>

<evaluation_criteria>
- Available tools must be fully and appropriately leveraged to meet the requirements.
- Tool call names must be valid, correct, and complete.
- Tool call arguments must be valid, correct, and complete.
- Fabrication, including the creation of information or knowledge not provided by the user, conflicting with user input, or not derived from the tools, must be penalized.
- Repetitive or unnecessary tool calls must be penalized.
- Excessive or unnecessary requests for user clarification beyond what is essential must be penalized.
</evaluation_criteria>

<conversation_history>
{chat_history}
</conversation_history>

<current_response_1>
{assistant_response_1}
</current_response_1>

<current_response_2>
{assistant_response_2}
</current_response_2>

Output your evaluation within <evaluation></evaluation> XML tags, and then enclose your choice (either '1' or '2') within <choice></choice> XML tags. Answer in the following format.
<evaluation>
{{your_evaluation}}
</evaluation>
<choice>
{{your_choice}}
</choice>

Figure 8: Evaluator prompt template of the pairwise critique task for non-reasoning LLMs.

---

**Serve as A Judge for Best-of-N Sampling (Think Mode)**

<task>
You are an expert evaluator of AI assistant performance. Given a complete user-assistant conversation history and {N} generated assistant responses, you are to conduct a thorough, fact-based, and comprehensive comparison. Based on specific evidence from your evaluation, make a clear choice of which response is superior. If multiple responses are identical and equally best, select the one with the smallest number.
</task>

<evaluation_criteria>
- Available tools must be fully and appropriately leveraged to meet the requirements.
- Tool call names must be valid, correct, and complete.
- Tool call arguments must be valid, correct, and complete.
- Fabrication, including the creation of information or knowledge not provided by the user, conflicting with user input, or not derived from the tools, must be penalized.
- Repetitive or unnecessary tool calls must be penalized.
- Excessive or unnecessary requests for user clarification beyond what is essential must be penalized.
</evaluation_criteria>

<conversation_history>
{chat_history}
</conversation_history>

<current_response_1>
{assistant_response_1}
</current_response_1>

<current_response_2>
{assistant_response_2}
</current_response_2>

…
<current_response_{N}>
{assistant_response_N}
</current_response_{N}>

Output your choice (a number between 1 and {N}) within <choice></choice> XML tags. No explanations should precede or follow the choice. Answer in the following format.
<choice>
{{your_choice}}
</choice>

Figure 9: Judge prompt template of the Best-of-N sampling task for reasoning LLMs.

---

**Serve as a Critic for Self-Correction (Think Mode)**

<task>
You are an expert evaluator of AI assistant performance. Given a complete user-assistant conversation history and a generated assistant response, you are to conduct a thorough, fact-based, and comprehensive evaluation. Based on specific evidence from your evaluation, provide a concise critique on how the current assistant response should be revised. If the response is entirely correct and requires no changes, output '[correct]' as your critique.
</task>

<evaluation_criteria>
- Available tools must be fully and appropriately leveraged to meet the requirements.
- Tool call names must be valid, correct, and complete.
- Tool call arguments must be valid, correct, and complete.
- Fabrication, including the creation of information or knowledge not provided by the user, conflicting with user input, or not derived from the tools, must be penalized.
- Repetitive or unnecessary tool calls must be penalized.
- Excessive or unnecessary requests for user clarification beyond what is essential must be penalized.
</evaluation_criteria>

<conversation_history>
{chat_history}
</conversation_history>

<current_response>
{assistant_response}
</current_response>

Output your final critique within <critique></critique> XML tags. No explanations should precede or follow the critique. Answer in the following format.
<critique>
{{your_critique}}
</critique>

---

Figure 10: Critic prompt template of the self-correction task for reasoning LLMs.

---

**Serve as an Editor for Self-Correction (No Think Mode)**

<task>
You are an expert editor of AI assistant response. Given a complete user-assistant conversation history, a generated assistant response, and a critique about how to improve it, your task is to produce the revised response.
</task>

<conversation_history>
{chat_history}
</conversation_history>

<current_response>
{assistant_response}
</current_response>

<critique>
{critique}
</critique>

Output the revised response within <revised_response></revised_response> XML tags. No explanations should precede or follow the response. Answer in the following format.
<revised_response>
{{revised_response}}
</revised_response>

---

Figure 11: Editor prompt template of the self-correction task for non-reasoning LLMs.

---

**System Prompt in Tool-Use Trajectory**

# Tools
You may call one or more functions to assist with the user query.
You are provided with function signatures within <tools></tools> XML tags:
<tools>
{tool_descs}
</tools>

For each function call, return a json object with function name and arguments within
<tool_call></tool_call> XML tags:
<tool_call>
{{"name": <function-name>, "arguments": <args-json-object>}}
</tool_call>

# Agent Policy
{agent_policy}

---

Figure 12: System prompt template of the tool-use trajectory.

# H  CASE STUDIES

**Valid Cases**  Through representative valid cases, we compare critiques from Claude 4 Sonnet and ToolRM-Qwen3-4B-Thinking-2507 on TRBench$_{\text{BFCL}}$ test samples. In the case shown in Figure 13, ToolRM accurately distinguishes correct from incorrect tool-call parameters without inducing "over-thinking" hallucinations when the user query plausibly maps to multiple candidate parameters. Another case in Figure 14 further demonstrates its tendency to ground analysis in contextual rationale rather than engage in speculative, divergent reasoning. Moreover, as shown in Figure 15, ToolRM adheres more closely to the evaluation criteria, preferring tool calls without redundant parameters. Taken together, these behaviors enable ToolRM to deliver reliable critiques in tool-use scenarios.

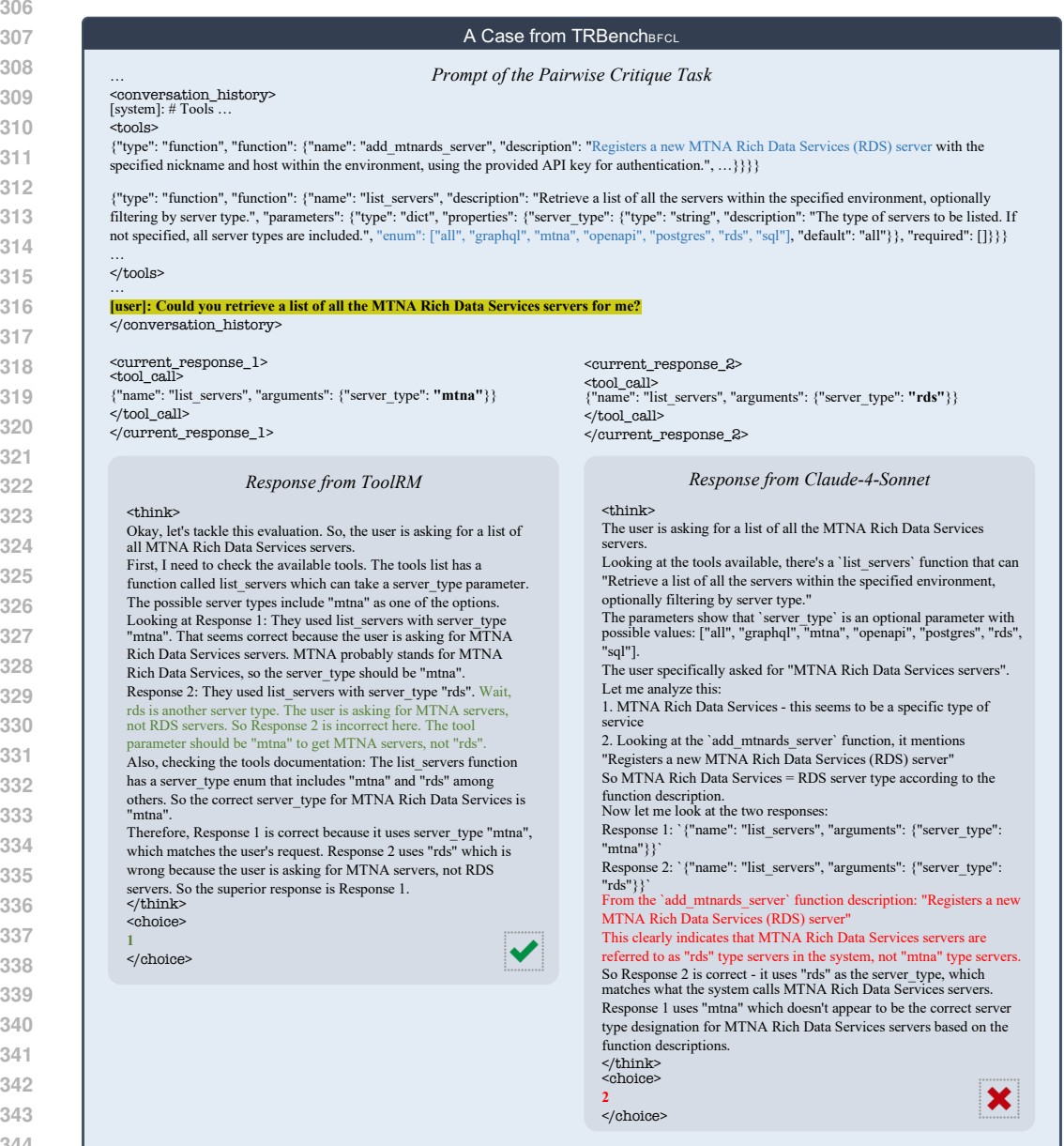

Figure 13: Valid *case 1* from TRBench$_{\text{BFCL}}$, featuring responses from ToolRM and Claude 4 Sonnet.

## A Case from TRBench_BFCL

### Prompt of the Pairwise Critique Task

...
`<conversation_history>`
[system]: # Tools ...
`<tools>`
{"type": "function", "function": {"name": "CustomDashboardsApi.get_custom_dashboards", "description": "Retrieves a list of custom dashboards that the user has access to, based on the provided query parameters.", "parameters": {"type": "dict", "required": ["query"], "properties": {"query": {"type": "string", "description": "A search query string to filter accessible dashboards. Supports logical operators and field-specific filtering."}, "pageSize": {"type": "integer", "description": "The number of dashboard entries to return per page.", "default": 10}, "page": {"type": "integer", "description": "The page number to retrieve, used for paginating through large sets of dashboard entries.", "default": 1}}}}}

{"type": "function", "function": {"name": "CustomDashboardsApi.get_custom_dashboard", "description": "Retrieve the details of a custom dashboard by its unique identifier.", "parameters": {"type": "dict", "required": ["customDashboardId"], "properties": {"customDashboardId": {"type": "string", "description": "The unique identifier of the custom dashboard to retrieve."}}}}}
...
`</tools>`
...
**[user]: Get the dashboard for "dashing".**
`</conversation_history>`

`<current_response_1>`
`<tool_call>`
{"name": "**CustomDashboardsApi.get_custom_dashboards**", "arguments": {**"query"**: "dashing"}}
`</tool_call>`
`</current_response_1>`

`<current_response_2>`
`<tool_call>`
{"name": "**CustomDashboardsApi.get_custom_dashboard**", "arguments": {**"customDashboardId"**: "dashing"}}
`</tool_call>`
`</current_response_2>`

### Response from ToolRM

`<think>`
Okay, let's tackle this evaluation. So, the user wants to get the dashboard for "dashing". The available tools are three functions: add_custom_dashboard, get_custom_dashboards, and get_custom_dashboard.
First, looking at the conversation history, the user's query is "Get the dashboard for 'dashing'". The assistant needs to use the tools correctly.
Looking at Response 1: They used get_custom_dashboards with query "dashing". That seems right because the get_custom_dashboards function takes a query parameter to find dashboards. The parameters are correct here.
Response 2: They used get_custom_dashboard with customDashboardId "dashing". Wait, the get_custom_dashboard function requires a customDashboardId, which is a unique identifier. But the user's query is "Get the dashboard for 'dashing'", which might mean they want a dashboard named "dashing" or with a title containing "dashing". However, the tool get_custom_dashboard needs a specific ID, not a name. The problem is that the user didn't provide an ID, so the assistant can't know the ID beforehand. Therefore, using get_custom_dashboard with ID "dashing" might be incorrect because the ID is a unique identifier, not the dashboard name. The correct approach would be to first use get_custom_dashboards to find the dashboard by name (query "dashing") and then get its ID to use in get_custom_dashboard. But in this case, the user's request is to get the dashboard for "dashing", so maybe they mean a dashboard named "dashing".
Wait, the tools description says for get_custom_dashboards, the query parameter is a search string. So Response 1 is correct because they're searching for dashboards with query "dashing".
Response 2 is trying to get a dashboard by ID "dashing", but the ID is a unique identifier that isn't necessarily the same as the name. Since the user said "dashboard for 'dashing'", it's more likely they want a dashboard whose name is "dashing" or contains "dashing". So the correct tool here is get_custom_dashboards with query "dashing" (Response 1). Response 2 is using get_custom_dashboard with an ID that's probably not valid because the ID is a unique identifier, not a name. The tool get_custom_dashboard requires a specific ID, which the user didn't provide. So Response 2 is incorrect because they're assuming "dashing" is a valid ID, but the tool's parameters need a real ID. Therefore, Response 1 is superior.
`</think>`
`<choice>`
**1**
`</choice>`

### Response from Claude-4-Sonnet

`<think>`
Let me analyze both responses based on the evaluation criteria.
The user is asking: "Get the dashboard for 'dashing'."
I need to evaluate which tool call is more appropriate.
Response 1 uses `CustomDashboardsApi.get_custom_dashboards` with arguments `{"query": "dashing"}`
Response 2 uses `CustomDashboardsApi.get_custom_dashboard` with arguments `{"customDashboardId": "dashing"}`
Let me look at the function signatures:
1. `CustomDashboardsApi.get_custom_dashboards`:
   - Description: "Retrieves a list of custom dashboards that the user has access to, based on provided query parameters."
   - Required parameters: ["query"]
   - This function searches for dashboards using a query string
2. `CustomDashboardsApi.get_custom_dashboard`:
   - Description: "Retrieve the details of a custom dashboard by its unique identifier."
   - Required parameters: ["customDashboardId"]
   - This function retrieves a specific dashboard by its ID
Now, the user said "Get the dashboard for 'dashing'". The key question is: is "dashing" a dashboard ID or a search term?
From the context, it's ambiguous. "Dashing" could be:
1. The unique identifier of a specific dashboard
2. A name or part of a name to search for
However, looking at the function descriptions:
- `get_custom_dashboard` is for retrieving a specific dashboard by its unique identifier
- `get_custom_dashboards` is for searching/filtering dashboards
The user said "Get THE dashboard for 'dashing'" - the use of "THE" suggests they're looking for a specific dashboard, not searching for multiple dashboards.
If "dashing" is indeed the dashboard ID, then Response 2 would be correct.
If "dashing" is a search term or dashboard name (not ID), then Response 1 would be correct.
The singular phrasing "the dashboard" implies a specific, unique dashboard. This strongly points to using the `get_custom_dashboard` method with "dashing" as the identifier.
I'll prioritize the most precise interpretation based on the user's language, which suggests a direct retrieval of a specific dashboard. The approach balances efficiency and user intent, targeting a direct dashboard fetch while maintaining flexibility if the identifier proves invalid.
The definite article implies a precise dashboard selection, making the direct retrieval method most aligned with the user's expectation. Searching becomes a secondary strategy if direct identification fails.
`</think>`
`<choice>`
**2**
`</choice>`

Figure 14: Valid *case 2* from TRBench_BFCL, featuring responses from ToolRM and Claude 4 Sonnet.

## A Case from TRBench_BFCL

*Prompt of the Pairwise Critique Task*

…
```
<conversation_history>
[system]: # Tools …
<tools>
{"type": "function", "function": {"name": "get_shortest_driving_distance", "description": "Calculate the shortest driving distance between two locations.",
"parameters": {"type": "dict", "properties": {"origin": {"type": "string", "description": "Starting point of the journey."}, "destination": {"type": "string",
"description": "End point of the journey."}, "unit": {"type": "string", "description": "Preferred unit of distance (optional, default is kilometers)."}},
"required": ["origin", "destination"]}}}
…
</tools>
…
[user]: Find the shortest driving distance between New York City and Washington D.C.
</conversation_history>
```

```
<current_response_1>                          <current_response_2>
<tool_call>                                   <tool_call>
{"name": "get_shortest_driving_distance",     {"name": "get_shortest_driving_distance",
"arguments": {"origin":                        "arguments": {"origin":
"New York City", "destination": "Washington D.C."}}   "New York City", "destination": "Washington D.C.", "unit": "miles"}}
</tool_call>                                   </tool_call>
</current_response_1>                          </current_response_2>
```

### Response from ToolRM

```
<think>
Okay, let's tackle this evaluation. So, the user wants the shortest
driving distance between New York City and Washington D.C.
The available tools include get_shortest_driving_distance, which
requires origin and destination. The other parameters have a unit
option, defaulting to kilometers.

Looking at Response 1: They called the function with origin and
destination, no unit specified. That's correct because the tool's
parameters say unit is optional with default kilometers. So that's
valid.

Response 2: They added "unit": "miles". The tool's description
says the unit is optional with default kilometers. So specifying
miles is okay, but does the tool accept that? The tool's parameters
say unit is optional, so it's valid. However, the user's query doesn't
specify a unit, so the assistant should probably use the default.
The tool's description says the default is kilometers, so maybe the
assistant shouldn't change it unless needed. Since the user didn't
ask for miles, Response 2 is adding an unnecessary parameter.
That would be a redundancy. The evaluation criteria say to avoid
unnecessary tool calls or parameters. So Response 2 has an extra
parameter that's not needed, which is a drawback. Therefore,
Response 1 is better because it doesn't add unnecessary info. The
tool's parameters say unit is optional, but since the user didn't
specify, the default is fine. So the superior response is Response 1.
</think>
<choice>
1
</choice>
```
✅

### Response from Claude-4-Sonnet

```
<think>
Let me analyze these two responses based on the evaluation criteria:
1. **Available tools must be fully and appropriately leveraged to
meet the requirements.**
Both responses use the correct tool `get_shortest_driving_distance`
which is appropriate for finding the shortest driving distance
between two locations.
2. **Tool call names must be valid, correct, and complete.**
Both responses use the correct function name
"get_shortest_driving_distance".
3. **Tool call arguments must be valid, correct, and complete.**
Let me check the function signature:
```
```
"parameters": {"type": "dict", "properties": {"origin": {"type":
"string", "description": "Starting point of the journey."},
"destination": {"type": "string", "description": "End point of the
journey."}, "unit": {"type": "string", "description": "Preferred unit
of distance (optional, default is kilometers)."}}, "required":
["origin", "destination"]}
```
```
Response 1 arguments:
- "origin": "New York City" ✓
- "destination": "Washington D.C." ✓
- Missing "unit" parameter (but it's optional)

Response 2 arguments:
- "origin": "New York City" ✓
- "destination": "Washington D.C." ✓
- "unit": "miles" ✓
Both responses provide the required parameters. Response 2 also
includes the optional "unit" parameter.
4. **Fabrication, including the creation of information or
knowledge not provided by the user, conflicting with user input, or
not derived from the tools, must be penalized.**
Neither response fabricates information. Both are making
appropriate tool calls.
5. **Repetitive or unnecessary tool calls must be penalized.**
Neither response makes repetitive or unnecessary calls.
6. **Excessive or unnecessary requests for user clarification beyond
what is essential must be penalized.**
Neither response requests clarification.
The key difference is that Response 2 includes the "unit" parameter
set to "miles". Since the user asked for the distance between two US
cities (New York City and Washington D.C.), it would be more
natural and user-friendly to provide the result in miles rather than
the default kilometers. While the "unit" parameter is optional,
specifying "miles" would likely give a more useful result for
someone asking about distances between US cities.
Response 2 demonstrates enhanced user-centricity by selecting miles
as the distance unit, which aligns more naturally with US
geographic conventions and user expectations.
</think>
<choice>
2
</choice>
```
❌

Figure 15: Valid *case 3* from TRBench_BFCL, featuring responses from ToolRM and Claude 4 Sonnet.

**Error Cases** We also present representative error cases of ToolRM on TRBench_{BFCL}, as discussed in Section 3.6. Figure 16 illustrates an error of type (i), where the model fails to reason correctly given an underspecified tool parameter description, while Figure 17 illustrates an error of type (ii), where the model fails to distinguish primary errors from secondary ones.

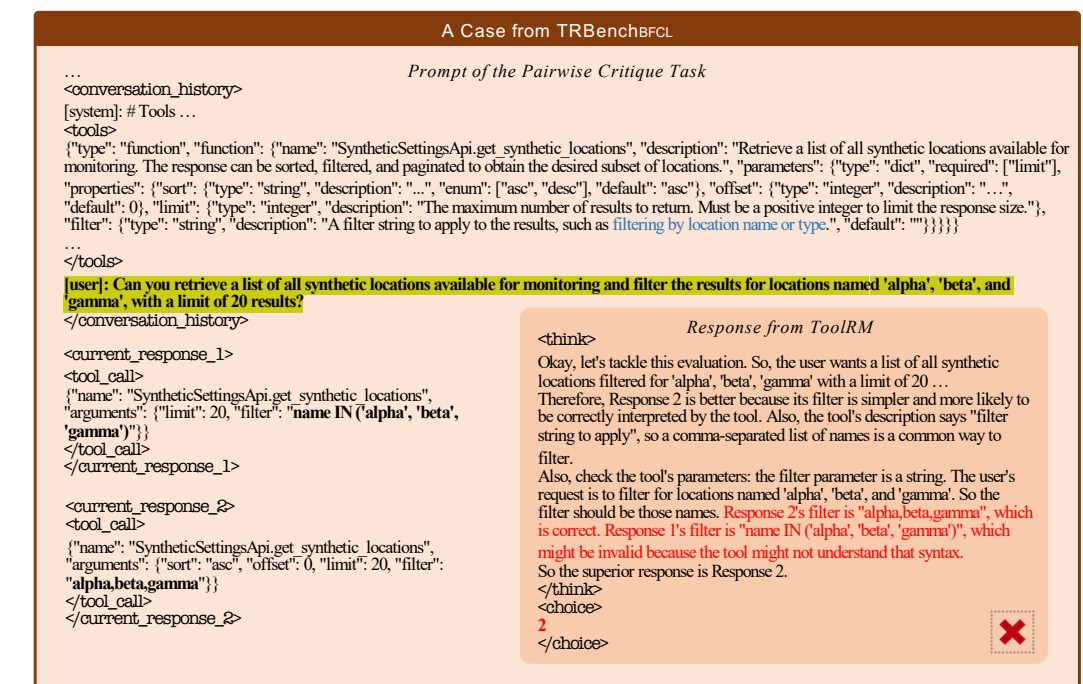

Figure 16: Error *case 1* from TRBench_{BFCL}, featuring responses from ToolRM.

**A Case from TRBenchBFCL**

*Prompt of the Pairwise Critique Task*

…
<conversation_history>
[system]: # Tools …
<tools>
{"type": "function", "function": {"name": "send_message", "description": "Sends a message to a human user, supporting a full range of Unicode characters, including emojis.", "parameters": {"type": "dict", "properties": {"message": {"type": "string", "description": "The content of the message to be sent. Supports Unicode characters such as text and emojis."}, "recipient": {"type": "string", "description": "The identifier of the recipient. …", "default": "default@example.com"}, …, "required": ["message"]}}}}

{"type": "function", "function": {"name": "recall_memory_search", "description": "Searches the prior conversation history for occurrences of a specified string and returns paginated results.", "parameters": {"type": "dict", "required": ["query"], "properties": {"query": {"type": "string", "description": "The search string to locate within the conversation history."}, "page": {"type": "integer", "description": "Specifies the page number of results to retrieve, where 0 represents the first page.", "default": 0}, "request_heartbeat": {"type": "boolean", "description": "When set to true, triggers an immediate heartbeat post function execution to facilitate follow-up actions.", "default": false}}}}}
…
</tools>
**[user]: Could you please find out when Shishir's birthday is from our previous conversations and then congratulate him by sending a 'Happy Birthday!' message regardless?**
</conversation_history>

<current_response_1>
<tool_call>
{"name": "recall_memory_search", "arguments": {"query": "Shishir Birthday"}}
</tool_call>
<tool_call>
**{"name": "send_message", "arguments": {"message": "Happy Birthday!", "recipient": "Shishir"}}**
</tool_call>
</current_response_1>

<current_response_2>
<tool_call>
{"name": "recall_memory_search", "arguments": {"query": "Shishir's birthday", "page": 0, "request_heartbeat": false}}
</tool_call>
</current_response_2>

*Response from ToolRM*

<think>
Okay, let's tackle this evaluation …
Wait, the tool recall_memory_search will return the birthday date, but the assistant then sends the message regardless. So maybe the send_message is done without waiting for the tool response? But according to the process, the assistant does tool call -> wait for response -> then next step. So Response 1 has two tool calls: first recall_memory_search, then send_message. That's correct.
But the key is that the recall_memory_search must be done with the correct query. Since the user said "Shishir's birthday", Response 2's query is more precise. So Response 2 is superior …
Given that, I think Response 2 is better because the query is more precise. So the choice is 2.
</think>
<choice>
2
</choice>

Figure 17: Error *case 2* from TRBenchBFCL, featuring responses from ToolRM.