# OpenReview forum: "One Model to Critique Them All: Rewarding Agentic Tool-Use via Efficient Reasoning"
_ICLR.cc/2026/Conference — ICLR 2026 Conference Withdrawn Submission_

### Official Review · Reviewer_MrPn · 2025-10-31

**Soundness:** 3
**Presentation:** 2
**Contribution:** 2
**Rating:** 6
**Confidence:** 4

**Summary:**

This paper introduces ToolRM, a family of lightweight generative reward models for LLM tool use. The authors propose a two-stage data construction pipeline that yields ToolPref-Pairwise-30K, a pairwise preference dataset created via rule-based scoring and multidimensional sampling. The paper also presents TRBENCH$_{\mathrm{BFCL}}$, a new benchmark for evaluating tool-use reward models. Extensive experiments demonstrate that ToolRM-trained models outperform strong baselines, including leading foundation and specialized models on pairwise reward judgment, scaling, and self-correction tasks.

**Strengths:**

- The paper addresses the clear bottleneck in reward model development for agentic tool-use with LLMs.
- The pipeline for ToolPref-Pairwise-30K is carefully devised. I checked some data samples and found them of high quality.
- TRBENCH$_{\mathrm{BFCL}}$ provides an OOD, error-rich environment for fair evaluation. This could potentially be a valuable asset for the tool-use RM community.

**Weaknesses:**

- The central algorithmic innovations are mostly in the data construction and evaluation pipeline, whereas the reward modeling itself leverages the well-established GRPO methods. No other significant technical contributions on the reward model side (e.g, architectural novelty) are shown.
- The distillation data construction pipeline may nevertheless introduce subtle biases or data artifacts that could be exploited by reward models. For instance, does the skew of preference intensity bins or complexity scores create blind spots? Seems the paper does not analyze or discuss the risk of overfitting to rule-based artifacts instead of learning genuine task competence.
- More error analysis would strengthen the paper. For example, do the models ever prefer responses that "look correct" but are subtly wrong, or exhibit brittle reasoning?

**Questions:**

Please see weaknesses.

---

> ### Author Response · Authors · 2025-11-22
>
> We sincerely appreciate the time you devoted to reviewing our work. Below, we respond to your questions one by one.
>
> ---
>
> **Q1: Novelty of this work**
> **A1:** Thank you for your positive assessment of our work. We acknowledge that the primary innovations lie in the construction of domain-specific data and the design of the evaluation pipeline, rather than in the RM training paradigm itself. We would also like to emphasize the key insight of this study: in agentic tool use scenarios, preference data generated through rule-based comparisons can, through RL, guide the RM to develop strong and generalizable factual reasoning abilities. This yields a scalable data-construction approach for this domain that does not rely on manual annotation.
>
> ---
>
> **Q2: Potential preference bias introduced by data distillation**
> **A2:** We have carefully considered the potential preference bias introduced by data distillation and have minimized this risk through the proposed data construction process:
> (i) In the reject response sampling phase, we adopt five models from three different model families, which helps reduce the risk of introducing the output distribution of any specific model into the following reward modeling.
> (ii) In the preference data selection phase, we explicitly count for three key dimensions that affect preference modeling: diversity of data sources, coverage of preference intensity, and task complexity. The designed BMDS strategy balances the distributions of both data sources and preference differences, thereby mitigating the risk of introducing data-centric blind spots into the trained RMs.
>
> In addition, based on prior work [1][2], selecting more complex tasks is beneficial for training: the ability to solve complex tasks tends to generalize effectively to simpler tasks and thus does not introduce subtle biases into the RM. Empirically, our experiments on TRBench-BFCL (see Table 2) demonstrate that ToolRM possesses genuine generalization capability. Although ToolRM is optimized according to accuracy rules for single-step function calls, it also achieves significant improvements on state-verification-based multi-turn tasks. This indicates that, through training, the model truly generalizes factual reasoning abilities across different tasks, rather than merely overfitting to the rules used in data construction.
>
> ---
>
> **Q3: Error analysis of ToolRM**
> **A3:** This is a very constructive suggestion. We further analyze the thinking process of ToolRM in cases where its final judgments are inconsistent with the ground-truth preferences. Our investigation indicates that these errors primarily fall into two categories:
> (i) When the description of a target tool’s functionality or parameters lacks concrete examples, the model is unable to infer the most appropriate tool invocation from the candidates, given the available tool information and the user’s query.
> (ii) The originally annotated chosen response contains minor errors, while the rejected response has more fundamental and severe errors. The model correctly identifies all errors but fails to distinguish primary errors from secondary ones, leading to an incorrect pairwise reward.
> We believe the first type of error is constrained by the base model’s inherent reasoning capability and is therefore more difficult to improve. The second type, however, is more tractable and can be mitigated through targeted optimization using higher-quality, non-perfect preference data—that is, cases where the chosen response still contains minor errors.
> Concrete TRBench-BFCL examples corresponding to these two error types are provided in the appendix for detailed reference. We hope it can address your concerns.
>
> ---
>
> We have added the additional experiments to the paper PDF. Please refer to it for details. We hope these responses address your concerns, and we look forward to any further feedback you may have.
>
> ---
>
> Reference:
> [1] Xu et al., WizardLM: Empowering large pre-trained language models to follow complex instructions, ICLR 2024.
> [2] Liu et al., What makes good data for alignment? a comprehensive study of automatic data selection in instruction tuning, ICLR 2024.

---

### Official Review · Reviewer_Ft4k · 2025-11-01

**Soundness:** 3
**Presentation:** 3
**Contribution:** 2
**Rating:** 4
**Confidence:** 4

**Summary:**

In this paper, the authors look at improving reward modeling capabilities for tools using responses. To do so, the authors propose a pipeline for generating preference data from tool use demonstrations. Based on the dataset generated from their pipeline, the authors train a series of ToolRM models which have improved tool critiquing capabilities. In addition, the authors propose a new tool use preference evaluation suite which they use to benchmark different methods.

**Strengths:**

* With the growing importance of agnetic tool using AI, collecting preference data in such settings seems very useful and like a directionally important research direction.
* The paper introduces a tool use preference eval which is a useful contribution as a benchmark for the RM research community to hill climb against.

**Weaknesses:**

* The ablations section does not ablate parameters at a useful granularity. There are many different factors that contribute to the BMDS algorithm, however, BMDS is treated as a monolithic block without decomposing which dimension matters most (source diversity, preference intensity, or task complexity)
* The performance between the base and fine-tuned llm judges in Table 2 doesn’t seem to be a fair comparison as the base models aren’t trained to be judges. To understand how useful the tool use preference data is, it would be useful to include a baseline of training the models to be judges on normal preference data. Without doing such it doesn’t seem possible to get a calibrated understanding of how useful the tool use preference data is.

**Questions:**

* Did the authors look at training non reasoning reward models on their task (i.e. standard single forward pass reward models [1]. It would be helpful to understand how useful reasoning is for verifying tool usage correctness.
* Similar to the second weakness, but for models not trained on the Hermes tool usage format there is a distribution shift when evaluating responses with such a format. It would be a useful comparison to simple SFT the base models on the preferred samples from the formatted tool use dataset and see how much of the gap is recovered.

[1] Ouyang, Long, et al. "Training language models to follow instructions with human feedback." Advances in neural information processing systems 35 (2022): 27730-27744.

---

> ### Author Response · Authors · 2025-11-22
>
> We sincerely appreciate the time you spent reviewing our work. Below we respond to each of your questions in turn:
>
> **Q1: Lack of fine-grained ablation on data selection algorithm**
> **A1:** Your comment is very helpful. We have added more fine-grained ablation results for BMDS: specifically, we individually ablate each of the three dimensions, retrain the model, and evaluate it on TRBench-BFCL. The results are shown below:
>
> | Model             | Avg. Acc | W-Avg. Acc      |
> |-------------------|---------:|----------------:|
> | Full ToolRM       | 79.77    | 71.87           |
> | - w/o Full BMDS   | 74.28    | 67.24 (-4.63)   |
> | - w/o DDS         | 76.21    | 68.64 (-3.23)   |
> | - w/o CPI         | 76.90    | 70.29 (-1.58)   |
> | - w/o CT          | 76.33    | 68.89 (-2.98)   |
>
> According to the results, each BMDS dimension contributes to final performance. Moreover, the diversity of data sources and task complexity have larger impacts than preference intensity, highlighting the importance of both diversity and contextual complexity for reward-model training.
>
> **Q2: Impact of data domain on ToolRM**
> **A2:** Thank you for carefully reading our paper. This suggestion is very helpful: adding a baseline trained on generic preference data makes the effectiveness of the domain-specific ToolPref dataset much clearer by comparison. To validate this point, we conduct the following experiments:
> (i) We randomly sampled 30,000 instances from Skywork-Reward-Preference-80K-v0.2 [1], a high-quality general preference dataset, matching the size of ToolPref-Pairwise.
> (ii) We made minimal modifications to the ToolRM prompt template (removing the detailed tool-use evaluation criteria) and used it to perform RL training on the baseline models in the same way as for previous ToolRM.
> (iii) The trained models were then evaluated on TRBench-BFCL with the same prompt. The results are shown below:
>
> | Model                                | W-Avg. Acc       |
> |--------------------------------------|------------------|
> | *Qwen3-4B-Instruct-2507*             | 59.67            |
> | - GenRM on NormalPref                | 63.82 (+4.15)    |
> | - GenRM on ToolPref                  | 66.85 (+7.18)    |
> | *Qwen3-4B-Thinking-2507*             | 57.59            |
> | - GenRM on NormalPref                | 63.19 (+5.60)    |
> | - GenRM on ToolPref                  | 71.87 (+14.28)   |
>
> According	to the results, models trained on high-quality normal preference data do improve their judging performance on pairwise classification tasks in the tool-use domain. However, our domain-specific preference dataset ToolPref delivers substantially larger gains over the base model, especially when training from a ‘thinking’ model, where RL objective fully exploits its potential for domain-specific reasoning.

---

> ### Author Response · Authors · 2025-11-22
>
> **Q3: Impact of training objective on ToolRM**
> **A3:** Thank you for the insightful suggestion. We are likewise interested in understanding the effect of training on ToolPref with a Bradley–Terry (BT) objective. To investigate this, we conduct additional experiments in which we train a ScalarRM using the BT objective and compare it with a GenRM trained using an RL objective. We then evaluate both models on TRBench-BFCL, with the results summarized below:
>
> | Model                                | W-Avg. Acc       |
> |--------------------------------------|------------------|
> | *Qwen3-4B-Instruct-2507*             | 59.67            |
> | - GenRM on ToolPref                  | 66.85 (+7.18)    |
> | - ScalarRM on ToolPref                  | 77.61 (+17.94)   |
> | *Qwen3-4B-Thinking-2507*             | 57.59            |
> | - GenRM on ToolPref                  | 71.87 (+14.28)   |
> | - ScalarRM on ToolPref                  | 76.80 (+19.21)   |
>
> These results show that ToolPref data remains effective under a BT objective and can further improve RM performance compared with the pairwise RL objective in pairwise preference classification tasks. This is consistent with our previous findings on the Skywork-Critic (GenRM) / Reward (ScalarRM) model series (see Table 2): when using the same base model and training data, ScalarRM trained with a BT objective naturally produces more accurate relative scores than GenRM.
>
> We also observe that *instruct* models, which have more concise output patterns, are better suited for ScalarRM with a BT objective to output precise scores, whereas *thinking* models, which produce longer initial chain of thoughts and exhibit stronger exploration capability, are better suited for RL training to obtain GenRM with stronger analytical ability. In practice, we believe each training objective has its strengths and should be used flexibly according to the application scenario: use GenRM in scenarios that need critique feedback and interpretability (e.g., self-correction), and use ScalarRM in scenarios that only require accurate reward scores (e.g., RL training / BoN sampling).
>
> To verify this, we further evaluate ToolRM trained with a BT objective as a BoN sampling judge, following the experimental setup in Section 3.2, and compare its performance against the generative ToolRM. The table below reports accuracy on the ACEBench normal subset:
>
> | Model                                          | Accuracy |
> |-----------------------------------------------|----------|
> | *Qwen3-4B-Instruct-2507* (policy model)       | 63.4     |
> | – BoN-16 w/ ToolRM (GenRM)                       | 66.6 (+3.2)  |
> | – BoN-16 w/ ToolRM (ScalarRM)                       | 67.2 (+3.8)  |
> | – Self-correction w/ ToolRM (GenRM)             | 74.8 (+11.4) |
>
> These results indicate that although discriminative ToolRM performs better than generative ToolRM when serving as a BoN sampling judger, overall performance is still constrained by the capability of the policy model itself (i.e., the best result among multiple candidate samples). By contrast, generative ToolRM is capable of acting as a critic for self-correction, producing high-quality critiques for any single sample generated by the policy model and thereby substantially raising the upper bound of performance.
>
> ---
>
> **Q4: Potential distribution shift caused by format misalignment**
> **A4:** We acknowledge that format misalignment could indeed lead to underperformance of the baseline models. However, according to the official implementation [2], the Qwen3 series is natively compatible with hermes-style tool calling without any additional fine-tuning. As a result, there may not be a format-aware distribution shift between the trained ToolRMs and their corresponding baseline models. The performance gains observed in ToolRM are therefore genuine.
>
> ---
>
> We have added the additional experiments to the paper PDF. Please refer to it for details. We hope these responses address your concerns, and we look forward to any further feedback you may have.
>
> ---
>
> Reference:
> [1] https://huggingface.co/datasets/Skywork/Skywork-Reward-Preference-80K-v0.2
> [2] https://github.com/QwenLM/Qwen-Agent/blob/main/qwen_agent/llm/fncall_prompts/nous_fncall_prompt.py

---

### Official Review · Reviewer_usxe · 2025-11-01

**Soundness:** 3
**Presentation:** 3
**Contribution:** 3
**Rating:** 4
**Confidence:** 3

**Summary:**

This paper focuses on reward models for function-calling tasks and proposes ToolRM, a generative reward model designed for tool-call scenarios. The authors develop a pipeline for constructing pairwise preference data based on rule-based scoring and multidimensional sampling, generating the ToolPref-Pairwise-30K dataset for critique tasks. Additionally, to evaluate RMs, the authors introduce TRBench-BFCL. Experimental results demonstrate the effectiveness of the proposed approach.

**Strengths:**

1. Reward modeling for tool-calling scenarios is crucial, and this work addresses this area by proposing a comprehensive and robust pipeline for preference data construction. The creation of ToolPref-Pairwise-30K and the accompanying benchmark represents a substantial contribution to the field.
2. The data construction process incorporates multiple verification methods to ensure data quality and employs diverse sampling strategies to enhance data diversity and quality. This systematic approach is commendable.
3. The experimental results demonstrate significant superiority of the proposed method, substantially improving accuracy across multiple models. The scaling experiments and analysis of critique effectiveness provide valuable insights.

**Weaknesses:**

1. For tool-calling reasoning, while many scenarios can be constrained through rule-based outcome rewards, process rewards for multi-turn tool calls are equally important. However, the paper focuses exclusively on outcome rewards without addressing process rewards, which limits the comprehensiveness of the approach.
2. The experiments lack direct application of the reward model to downstream tasks. For instance, there is no evaluation of how the reward model enhances tool-calling agents when combined with reinforcement learning methods such as PPO or GRPO, which would provide stronger evidence of practical utility.
3. The title mentions "efficient reasoning," but it is unclear which specific aspect this refers to. Intuitively, using RMs to constrain agentic tool-use should introduce computational overhead and potentially slow down the overall process, making the efficiency claim questionable without proper justification.

**Questions:**

N/A

---

> ### Author Response · Authors · 2025-11-22
>
> We sincerely thank you for the time and effort you dedicated to reviewing our work. Below we respond to your questions one by one.
>
> **Q1: Limitation of rule-based ORM**
> **A1:** This is an important and insightful question. First, we would like to emphasize that the primary goal of this work is not to adapt the reward model to a fixed set of rules in the tool-use domain. Rather, we aim to leverage the accurate signals derived from these rules to guide the model’s exploration–exploitation behavior during RL, thereby strengthening its factual reasoning ability. Empirically, in the TRBench-BFCL experiments, ToolRM also achieves substantial gains on state-verification-based, multi-turn tasks (see the *MTB* column in Table 2), demonstrating strong generalization of its reasoning capabilities beyond the rule-specified setting.
>
> Second, regarding the categorization of ToolRM in agentic tool-use scenarios (ORM vs. PRM), we believe it depends on the application context:
>
> (i) *Single agent action perspective.* ToolRM is trained to evaluate the final output of each action (without access the intermediate chain-of-thoughts). Under this view, it functions as an outcome reward model.
> (ii) *Multi-turn agentic task perspective.* For longer trajectories within multi-turn tasks, ToolRM’s main role is to evaluate each agent step individually along the trajectory. It is not designed to assess the final task outcome (if any) or to provide holistic feedback on the entire trajectory. From this perspective, it aligns more with a process reward model.
>
> We agree that extending ToolRM to evaluate entire trajectories is both important and technically challenging. Since this would entail different application goals and substantial additional work, we leave it as a direction for future research.
>
> ---
>
> **Q2: Justification of ''efficient reasoning'' in ToolRM**
> **A2:** In this paper, we use the term ''efficient reasoning'' to describe the behavior of ToolRM when it acts as a critic to enhance the policy model’s performance at inference time. Compared with other generative RM baselines, ToolRM produces higher-quality critiques at lower inference cost; detailed evidence and analysis are provided in Section 3.3.
>
> Our self-correction results on ACEBench are summarized in Figure 3. Relative to the policy model performing single-pass inference without RM guidance (w/o Critic), introducing ToolRM does add inference cost, but yields a substantial performance improvement (+11.4). Compared with the baseline RM (w/Base)—which also incurs additional overhead—ToolRM not only generates more effective critiques, but also reduces the required reasoning tokens for those critiques by more than 66%.
>
> These results indicate that, in the context of critique-task training for agentic tool use, ToolRM has learned an efficient and reliable reasoning strategy, and that this strategy generalizes to out-of-distribution tasks in the same domain. This is why we highlight this property in the title, introduction, and conclusion, and we believe the experiments provide adequate evidence to justify the term ''efficient reasoning''.
>
> ---
>
> **Q3: Lack of downstream application of RM in policy model RL**
> **A3:** This comment is very constructive. We fully agree that using ToolRM feedback to train and evaluate the policy model on downstream tasks would offer a stronger and more direct demonstration of the benefits of our approach. However, incorporating the pairwise generative RM for agent training is relatively complex and requires substantial time and computational resources. Despite these challenges, we are actively working to conduct such experiments as soon as possible. We hope to update these results in a few days to further strengthen our claims.
>
> ---
>
> We hope these responses address your concerns, and we look forward to any further feedback you may have.

---

### Official Review · Reviewer_F85k · 2025-11-03

**Soundness:** 2
**Presentation:** 2
**Contribution:** 3
**Rating:** 2
**Confidence:** 5

**Summary:**

The paper presents a generative reward model for tool-use evaluation. The authors construct a synthetic pairwise preference dataset by verifying tool-call trajectories, sampling candidate model responses, and scoring them with a rule-based function. A balanced multi-dimensional sampling strategy selects challenging examples. The model is trained via GRPO, where each pairwise example is converted into a forced-choice evaluation task, and the model receives a binary reward depending on whether its <choice> tag matches the reference.

**Strengths:**

1. The paper shows useful engineering contributions. It constructs an LLM-generated tool-use preference dataset, although not compare with previous preference dataset generation pipeline.

2. It introduces a benchmark for tool-RM evaluation. The curated dataset considers diversity and preference density.

3. Experiments on classification accuracy include diverse LLMs/APIs.

**Weaknesses:**

1. **Missing positioning and comparison against existing tool-augmented reward modeling work**. The paper does not cite or compare with prior works that explicitly frame and study tool-augmented reward models (ICLR’24 and follow-up papers). These directly address reward modeling for tool agents, and omitting them weakens the novelty claim and contextualization.

2. Pairwise dataset but no pairwise optimization objective. Although the dataset contains pairwise preferences, the training uses binary RL reward instead of pairwise margin (e.g., Bradley-Terry, InfoNCE for prefs). This forfeits preference strength information and deviates from standard RM objectives. A comparison with classical pairwise RM training is highly required.

3. **Relies on SFT / thinking-tuned Qwen models, without ablation on initialization**. All experiments are initialized from instruction-tuned and thinking-tuned Qwen checkpoints. No experiment isolates the effect of RL vs. SFT initialization or evaluates purely pretrained baselines.

4. **Unclear presentation**. e.g., Table 2 uses symbols (S, M, P, PM, LS, LM, LP, LPM), but no definitions are given in main text. It is unclear what those symbols mean.

5. Limited failure analysis / error taxonomy. The paper would benefit from qualitative cases where TOOLRM fails or is brittle in long-horizon or multi-tool settings.

6. **Limited LLM family choice**. The paper only conducts experiments on Qwen models, instead of others. It is unclear about the generalization to other LLMs, considering that some phenomenon may only occur on Qwen [2,3].

7. **Lack of human evaluation or expert validation**. All preference labels are either rule-based or model-generated. No human annotator study or expert evaluation is provided.

The idea of using tool-use preferences to build generative RMs is beneficial, but conceptual novelty is modest relative to expanding prior work on tool-augmented RL, tool-aware critics, and RM-as-reasoning architectures.


**References**:

[1] Tool-augmented reward modeling. ICLR 2024.

[2] Spurious Rewards: Rethinking Training Signals in RLVR. arxiv 2025.

[3] Reasoning or Memorization? Unreliable Results of Reinforcement Learning Due to Data Contamination. arxiv 2025.

**Questions:**

1. I suggest the authors compare with and cite prior tool-augmented RM literature (ICLR’24 and successors).
What is the conceptual delta relative to those works?

2. Why choose binary RL reward over pairwise margin loss? Would a hybrid objective help?

3. Can you provide SFT-initialization ablation, especially compare with cold-started data?

4. What are the results on other LLMs? Can the reported conclusion generalize to other LLMs?

5. Could the authors provide a failure mode breakdown analysis? Also, the authors can share qualitative examples of incorrect reward assignments.

6. How sensitive are results to the rule-based scoring heuristics? Would human preference data outperform purely rule-based sampling?

7. Did the authors conduct any human or expert evaluation to verify whether ToolRM's preference decisions align with real human judgments for tool-use effectiveness and correctness?

---

> ### Author Response · Authors · 2025-11-22
>
> We sincerely thank you for the time and effort you dedicated to reviewing our work. Below we respond to your questions one by one.
>
> ---
>
> **Q1: Difference between this work and tool-augmented reward modeling?**
> **A1:** Thank you for the suggestion. The line of work on tool-augmented reward modeling is indeed one branch of reward modeling, and we will add the relevant citations in the paper. However, we emphasize that **our work and 'tool-augmented reward modeling' are conceptually different, with distinct motivations and goals**.
>
> Concretely, ToolRM does not invoke external tools during the inference process; instead, it is trained specific for evaluating tasks in *agentic tool-calling* settings. In contrast, the tool-augmented RM works you mentioned incorporate external tools during RM evaluation to improve the reliability of reward estimation, and they focus on tasks such as *general QA, writing, and coding*, where the policy model can complete the task without any tool calls. Those models are not trained to evaluate the tool-use trajectories of another policy model and therefore are not applicable to the scenario studied in this paper, nor can they be directly compared with our work.
>
> ---
>
> **Q2: Binary RL reward vs. pairwise margin loss**
> **A2:** This is a very valuable and interesting question. In this work, addition to scalar reward scores, we want the model to produce an analysis process (a critique) for each evaluation task. This improves the interpretability of the RM and enables self-correction for test-time scaling of the policy model (e.g., Section 3.3). Therefore, in this paper we primarily train ToolRM with a binary RL objective (GenRM), rather than the traditional pairwise margin loss (ScalarRM).
>
> We do evaluate a mixed-objective model, Cloud-RM, in the experiments (see Section 3.1), but developing a new RM training paradigm is not the main focus of this work.
>
> To further validate the effectiveness of ToolPref data, we additionally train models with the Bradley-Terry (BT) loss using the same base models and data, and evaluate them on TRBench-BFCL. Evaluation results are shown in the table below:
>
> | Model                                 | W-Avg. Acc    |
> |--------------------------------------|---------------|
> | *Qwen3-4B-Instruct-2507*             | 59.67         |
> | – GenRM on ToolPref                  | 66.85 (+7.18) |
> | – ScalarRM on ToolPref                  | 77.61 (+17.94)|
> | *Qwen3-4B-Thinking-2507*             | 57.59         |
> | – GenRM on ToolPref                  | 71.87 (+14.28)|
> | – ScalarRM on ToolPref                  | 76.80 (+19.21)|
>
> These results show that ToolPref data remains effective under a BT objective and can further improve RM performance compared with the pairwise RL objective in pairwise preference classification tasks. This is consistent with our previous findings on the Skywork-Critic (GenRM) / Reward (ScalarRM) model series (see Table 2): when using the same base model and training data, ScalarRM trained with a BT objective naturally produces more accurate relative scores than GenRM.
>
> We also observe that *instruct* models, which have more concise output patterns, are better suited for ScalarRM with a BT objective to output precise scores, whereas *thinking* models, which produce longer initial chain of thoughts and exhibit stronger exploration capability, are better suited for RL training to obtain GenRM with stronger analytical ability. In practice, we believe each training objective has its strengths and should be used flexibly according to the application scenario: use GenRM in scenarios that need critique feedback and interpretability (e.g., self-correction), and use ScalarRM in scenarios that only require accurate reward scores (e.g., RL training / BoN sampling).
>
> To verify this, we further evaluate ToolRM trained with a BT objective as a BoN sampling judge, following the experimental setup in Section 3.2, and compare its performance against the generative ToolRM. The table below reports accuracy on the ACEBench normal subset:
>
> | Model                                          | Accuracy |
> |-----------------------------------------------|----------|
> | *Qwen3-4B-Instruct-2507* (policy model)       | 63.4     |
> | – BoN-16 w/ Gen. ToolRM                       | 66.6 (+3.2)  |
> | – BoN-16 w/ Scalar. ToolRM                       | 67.2 (+3.8)  |
> | – Self-correction w/ Gen. ToolRM              | 74.8 (+11.4) |
>
> These results indicate that although discriminative ToolRM performs better than generative ToolRM when serving as a BoN sampling judger, overall performance is still constrained by the capability of the policy model itself (i.e., the best result among multiple candidate samples). By contrast, generative ToolRM is capable of acting as a critic for self-correction, producing high-quality critiques for any single sample generated by the policy model and thereby substantially raising the upper bound of performance.

---

> ### Author Response · Authors · 2025-11-22
>
> **Q3: Ablation on SFT initialization**
> **A3:** Based on extensive prior work in reward modeling, both ScalarRM (e.g., Skywork-Reward, InternLM-Reward) and GenRM (e.g., RM-R1, RRM) are typically trained on instruction-tuned models rather than on cold-start, purely pre-trained models. Instruction tuning enables models to better capture complex human preferences and task specifications. As a result, training a reward model directly from a purely pre-trained backbone is expected to be substantially more challenging to optimize. For this reason, we do not pursue such experiments in this work.
>
> ---
>
> **Q4: Unclear presentation in Table 2**
> **A4:** Due to page limits, we previously abbreviated the split names in BFCL. Following your suggestion, we have now added the full split names and their abbreviations in the main text.
>
> ---
>
> **Q5: Generalizability of ToolRM to different model families**
> **A5:** Thank you for the suggestion. We have added ToolRM trained on Llama-3 and xLAM-2 and report their evaluation results on TRBench-BFCL (see Table 2). The results show that ToolPref-Pairwise can significantly improve RM evaluation accuracy across different model families, validating the quality of the proposed dataset.
>
> ---
>
> **Q6: Analysis of model failure**
> **A6:** This is a very constructive suggestion. We further analyze the thinking process of ToolRM in cases where its final judgments are inconsistent with the ground-truth preferences. Our investigation indicates that these errors primarily fall into two categories:
> (i) When the description of a target tool’s functionality or parameters lacks concrete examples, the model is unable to infer the most appropriate tool invocation from the candidates, given the available tool information and the user’s query.
> (ii) The originally annotated chosen response contains minor errors, while the rejected response has more fundamental and severe errors. The model correctly identifies all errors but fails to distinguish primary errors from secondary ones, leading to an incorrect pairwise reward.
> We believe the first type of error is constrained by the base model’s inherent reasoning capability and is therefore more difficult to improve. The second type, however, is more tractable and can be mitigated through targeted optimization using higher-quality, non-perfect preference data—that is, cases where the chosen response still contains minor errors.
>
> ---
>
> **Q7:  Priority of human preference data and sensitivity of rule-based scoring**
> **A7:** Both data quality and diversity are crucial for reward model (RM) training. However, in real-world settings, it is prohibitively expensive to have human experts label large-scale tool-use preference data. To address this, we construct preference data from seven open-source tool-use datasets. Because the high-quality trajectories in these datasets have already been manually or automatically validated, we directly use them to extract the chosen responses in preference pairs, and then apply rule-based scoring to generate and identify the corresponding rejected responses.
>
> We argue that, given a fixed reference, our rule set is essentially an explicit abstraction of human comparative judgments. As such, it admits little scope for meaningful variations and thus is not truly “sensitive” in the sense of being fragile or arbitrary. Under this condition, the rules can be used directly for accurate preference labeling.
>
> ---
>
> **Q8: Lack of human evaluation**
> **A8:** In this paper, we evaluate RM effectiveness using two agentic tool-use benchmarks, BFCL and ACEBench. These datasets were carefully labeled and checked during construction to ensure answer correctness. Therefore, performance on these datasets directly reflects human preferences and the correctness of tool use, and no additional human evaluation is strictly necessary.
>
> ---
>
> **Finally, we would like to reiterate the importance of this work. As reviewers usxe, Ft4k, and MrPn noted, designing reward models specifically for agentic tool use—distinct from tool-augmented reward modeling—is a highly practical and important problem.** Our paper addresses a clear bottleneck in this area by introducing a novel, high-quality preference data construction pipeline that can support future scaling of both training and inference in this domain.
>
> We have added the additional experiments to the paper PDF. Please refer to it for details. We hope these responses address your concerns and we look forward to your further feedback.

---

### Note · Authors · 2025-12-02

I have read and agree with the venue's withdrawal policy on behalf of myself and my co-authors.